# Divergent evolutionary trajectories following speciation in two ectoparasitic honey bee mites

Maeva A. Techer [1]*, Rahul V. Rane[2,3], Miguel L. Grau[1], John M.K. Roberts [2], Shawn T. Sullivan[4], Ivan Liachko[4], Anna K. Childers [5], Jay D. Evans [5] & Alexander S. Mikheyev [1,6]*

Multispecies host-parasite evolution is common, but how parasites evolve after speciating remains poorly understood. Shared evolutionary history and physiology may propel species along similar evolutionary trajectories whereas pursuing different strategies can reduce competition. We test these scenarios in the economically important association between honey bees and ectoparasitic mites by sequencing the genomes of the sister mite species *Varroa destructor* and *Varroa jacobsoni*. These genomes were closely related, with 99.7% sequence identity. Among the 9,628 orthologous genes, 4.8% showed signs of positive selection in at least one species. Divergent selective trajectories were discovered in conserved chemosensory gene families (IGR, SNMP), and Halloween genes (CYP) involved in moulting and reproduction. However, there was little overlap in these gene sets and associated GO terms, indicating different selective regimes operating on each of the parasites. Based on our findings, we suggest that species-specific strategies may be needed to combat evolving parasite communities.

[1] Okinawa Institute of Science and Technology, 1919-1 Tancha Onna-son, 904-0495 Okinawa, Japan. [2] Commonwealth Scientific and Industrial Research Organisation, Clunies Ross St, (GPO Box 1700), Acton, ACT 2601, Australia. [3] Bio21 Institute, School of BioSciences, University of Melbourne, 30 Flemington Road, Parkville, VIC 3010, Australia. [4] Phase Genomics Inc, Seattle, WA 98195, USA. [5] USDA-ARS Bee Research Lab, Beltsville, MD, USA. [6] Australian National University, Canberra, ACT 2600, Australia. *email: maeva.techer@oist.jp; alexander.mikheyev@oist.jp

nteractions between hosts and parasites shape biodiversity by driving coevolutionary arms races and by regulating populations over ecological timescales. Parasitism is a remarkably successful strategy, occurring in nearly half of the recognized animal phyla[1,2]. Parasites depend on their hosts which form their principal ecological niches and provide essential resources to survive[3]. Commonly, multiple parasite generations elapse per host generation. Thus parasites are prone to evolve rapidly and may circumvent any gained host advantage[4]. The similar selective regimes acting on phylogenetically unrelated parasites resulted in reproducible evolution such as genome size reduction, functional losses associated with reduced metabolism, and adaptive functional gains linked to parasite virulence[5]. Most models of parasite coevolution are limited to interactions between single focal parasites and their hosts, in part because multispecies dynamics are much more complex[6,7]. In nature, the reality is that parasite's success and evolution will also depend on interactions with co-infecting conspecific and heterospecific parasites within its host niche[8–10]. Yet, coevolutionary dynamics involving more than one parasitic species remain poorly understood[11,12].

Co-infecting parasites occupying the same host can exclude competitors via direct, or interference competition[13,14]. One strategy to reduce interspecific competition can be spatial separation within regions of the host gut as adopted by cestodes that naturally co-infect sticklebacks[15]. Within the same host, spatial segregation that allows parasite coexistence can ultimately lead to intrahost speciation, as found in gill parasites (*Dactylogyrus*) of freshwater fish (*Cyprinidae*)[16]. Niches may also be partitioned temporally, which allows two fungal sibling species (*Erysiphe quercicola* and *E. alphitoides*) to exploit the same host tree (*Quercus robur*) by reducing direct interference[17]. Alternatively, parasites may shift their strategies to exploit different aspects of the host niche. For example, fire ants (*Solenopsis spp.*) are parasitized by a genus of decapitating phorid fly (*Pseudacteon*), which are highly host-specific and have the same life cycle, but specialize on attacking ants in different contexts (e.g., while foraging, vs. at the nest)[18]. However, almost nothing is known about evolutionary trajectories of speciating parasites. On one hand, they share physiology and host-specific adaptations, which may predispose them to evolve along common lines, particularly in allopatry. On the other hand, selective regimes may be less predictable, or even disruptive, in the case of character displacement.

We investigated the evolution of speciating parasites by sequencing genomes of two economically important mite species that specialize on honey bees (*Apis sp.*): *Varroa jacobsoni* and *Varroa destructor*. The honey bee colony, which is a densely packed community of genetically similar individuals, can host many diseases and parasites. Varroa mites are obligatory ectoparasites which feed on fat bodies of developing larvae/pupae and adult honey bees (shown in *V. destructor*)[19] and reproduce in the brood[20]. Obviously, host-feeding by Varroa mites injures vital organs of the honey bee, which can also lead to abnormal development[21] and to reduced lifespan[22]. In addition, Varroa mites are known to vector several viruses[23] and to transmit pathogenic bacteria[24]. These combined negative effects on individual honey bee decrease the fitness of the colony as a superorganism. Without acaricide treatments, managed western honey bee colonies were reported to decline and die within two to 3 years[20]. As its name indicates, *V. destructor* is considered as the main devastating force behind colony collapses and impact deeply western honey bee (*A. mellifera*) beekeeping[25,26].

The first record of Varroa mite parasitism dates back to 1904 in Java in where female mites were described from an Indo-Malayan *Apis cerana* honey bee colony (ex. *Apis indica*)[27]. Not until the mid-20th century did Varroa draw attention again when they were found to infest western honey bee *A. mellifera* colonies introduced in far eastern Asia[28]. Originally from Europe, Africa, and the Middle East, the western honey bee was allopatric to the rest of the Asian *Apis* species, and had no indigenous brood ectoparasites[29]. The host-switched mites rapidly spread worldwide and caused dramatic losses of *A. mellifera* colonies. They were identified as *Varroa jacobsoni* until the 2000s when a key taxonomic revision of the genus described them as a cryptic sister species *V. destructor*[30]. The taxonomic confusion existed and persisted for a while because both mites have similar life cycles and morphology[30] and both originally parasitized the widespread eastern honey bee (*Apis cerana*). The Varroa cryptic species complex was resolved by a combination of morphometrics and mitochondrial sequencing, which identified at least four mite species: *V. destructor*, *V. jacobsoni*, *V. rindereri*, and *V. underwoodi*[30].

As western honey bees were brought into contact with the eastern honey bee in Asia and Oceania, both *V. destructor* repeatedly colonized this novel host, followed by *V. jacobsoni* in 2008 in Papua New Guinea[28,31]. In contrast to the population collapses of *A. mellifera*, the host-parasite interaction with *A. cerana* is a balanced relationship in which a tolerance equilibrium is maintained by a range of parasite adaptations and host counter-adaptations[32–37]. Specifically, the eastern honey bee has impressive host defense traits, including (i) self and collective grooming behaviors[38], (ii) hygienic behavior where mites are removed from infested cell[39], (iii) worker brood death that stops parasite reproduction[34,35], (iv) entombing of mites in drone brood, and (v) shorter worker development cycle limiting the mites' offspring number[33]. By contrast, most western honey bee populations lack these defensive mechanisms, and hives collapse quickly after Varroa infestation. By contrast, Varroa mites have evolved highly specialized strategies to find, select, feed, reproduce, and hide on their host[40]. While many mite adaptations have already been discovered[20,40], the recent use of 'omics approaches has quickly broadened our understanding on *V. destructor* hidden specialization to host[37,41–43]. For example, transcriptomic and proteomic profiling of different *V. destructor* life stages boosted the identification of genomic regions related to chemosensory system[41,43,44], reproductive success[42], and linking oogenesis initiation or failure to parasitic reduced metabolism and host defense[45,46].

Here, we ask how both mite species evolved toward a tolerance equilibrium with *A. cerana*—did they follow similar or different evolutionary paths? On one hand, the two species naturally occur in parapatry, which is expected to result from divergent local adaptation, perhaps to the distribution of different *A. cerana* lineages[47,48] or to the climate. In actuality, what drives the parapatric distribution is unclear and recent evolutionary history shows that massive host-associated range shifts are possible. For instance, *V. destructor*, which was historically absent from Southeast Asia, now occurs there on its novel host *A. mellifera* in sympatry with *V. jacobsoni*, which does not parasitize *A. mellifera* outside of Melanesia. Thus, it is possible that the current geographical distribution resulted from secondary contact following allopatric speciation. This view is further supported by the fact that both species have the genetic capacity to shift to *A. mellifera* as a novel host, suggesting a level of physiological similarity. We tested these alternative hypotheses (parallel vs. divergent evolution) by generating high-quality hybrid de novo genome assembly of both species and examining them for signatures of adaptive evolution. We also compared them to the recently published genome of a distantly related ectoparasitic mite *Tropilaelaps mercedesae*, to see whether the patterns appear universal at different timescales. We found here that although the two Varroa species share 99.7% sequence identity across the nuclear genome

**Table 1 Statistics for the release of new genome assembly for honey bee *Varroa* mites and improvement from the first *V. destructor* draft genome**

| | *V. destructor* | | | *V. jacobsoni* |
|---|---|---|---|---|
| Genome assembly | BRL_Vdes_1.0 | Vdes2.0 | Vdes_3.0 | Vjacob_1.0 |
| NCBI accession | GCA_000181155.1 | GCA_000181155.2 | GCA_002443255.1 | GCA_002532875.1 |
| Assembly size (Mb) | 294.13 | 331.92 | 368.94 | 365.59 |
| Coverage | 5x | 60x | 119x | 57x |
| Gap size (bp) | 0 | 2,821,918 | 271,335 | 408,908 |
| GC content (%) | – | 40.1 | 40.9 | 40.9 |
| Scaffold | | | | |
| Scaffolds (n) | – | 20,448 | 1426 | 4881 |
| N50 scaffold length (bp) | – | 128,078 | 58,536,683 | 233,810 |
| L50 scaffold length (bp) | – | 703 | 3 | 482 |
| Contig | | | | |
| Contigs (n) | 184,190 | 52,152 | 4498 | 8241 |
| N50 contig length (bp) | 2262 | 15,568 | 201,886 | 96,009 |
| L50 contig length (bp) | 40,912 | 6465 | 556 | 1168 |
| Annotation | | | | |
| Gene number (n) | – | – | 12,849 | 15,486 |
| Mean length (bp) | – | – | 18,634 | 14,657 |
| mRNA number (n) | – | – | 30,208 | 26,214 |
| Mean mRNA length (bp) | – | – | 4193 | 3489 |
| CDS number (n) | – | – | 30,208 | 26,214 |
| Mean CDS length (bp) | – | – | 1953 | 1695 |
| Intron number (n) | – | – | 94,154 | 93,749 |
| Mean intron length (bp) | – | – | 3299 | 2898 |
| Exons number (n) | – | – | 110,06 | 111,624 |
| Mean exons length (bp) | – | – | 537 | 487 |
| Source | 49 | | This study | This study |

they have gone on largely dissimilar evolutionary trajectories after their split.

## Results

**Two new reference genomes for *V. destructor* and *V. jacobsoni*.** We took advantage of the haplodiploid sex determination of both *V. destructor* and *V. jacobsoni* to generate data to reduce the ambiguity due to heterozygous sites for a high contiguous hybrid genome assembly. The sequencing strategy adopted in this study combined shorts reads (Illumina) for which input DNA can be obtained from a single mite individual, long reads of tens of kilobases (PacBio), which requires a large quantity of high-molecular-weight DNA and Hi-C to capture chromosome conformation and interactions. We generated 28.6 Gb (333 ± S.D. 56 bp) and 56.2 Gb (409 ± S.D. 61 bp) of de-duplicated, decontaminated, and merged Illumina reads for *V. jacobsoni* and *V. destructor*, corresponding to a coverage of 80× and 152×, respectively. PacBio sequencing yielded 20.3 Gb of *V. destructor* reads with an N50 of 13.5 kb (66× coverage), which was used to fill 84.4% of the gaps in the scaffolded HiC assembly. Final de novo assemblies were named Vdes_3.0 (GCA_002443255.1) and Vjacob_1.0 (GCA_002532875.1) and are available on NCBI.

Through the k-mer analysis, the genome size estimates ranged from 369.02 to 365.13 Mbp using k = 42 (Supplementary Fig. 1). Both Vdes_3.0 and Vjacob_1.0 final assembly size were close to the predicted estimates but *V. jacobsoni* genome size was slightly shorter than *V. destructor* with 365.59 < 368.94 Mbp, respectively (Table 1). Previous assemblies size and contiguity for *V. destructor* with Vdes_1.0[49] and Vdes_2.0 were improved as detailed in Table 1. According to the N50 statistics, more than 50% of the genome was in scaffolds of 58.5 Mbp or more for Vdes_3.0 while the Vdes_2.0 N50 was around 0.1 Mbp. When comparing to invertebrate assemblies, Vdes_3.0 genome is among the most highly contiguous with to date the best-reported scaffold N50 and best

contig N50 for Acari and Mesostigmata as reported on the i5k database (i5k.github.io/arthropod_genomes_at_ncbi)[50].

Using Hi-C data, seven major scaffolds were constructed for *V. destructor* ranging from 39.4 to 76.9 Mbp with additional of 1,418 nuclear minor scaffolds > 0.2 Mbp. The longest genomic scaffolds correspond to the seven haploid chromosomes (2n = 14) previously detected in *V. destructor* karyotype[51]. Contrary to Vdes_3.0, which benefited from long reads and chromosome capture sequencing, *V. jacobsoni* Vjacob_1.0 genome was assembled from 8241 contigs into 4881 scaffolds with an N50 of 233,810 bp. Similar GC content composed 40.9% were found for both sister species Vdes_3.0 and Vjacob_1.0. The automated NCBI Eukaryote annotation pipeline predicted 10,247 protein-coding genes (out of 12,849 genes and pseudo-genes) for Vdes_3.0, and 10,739 protein-coding genes (out of 15,486 genes and pseudo-genes) for Vjacob_1.0[52] (based upon transcriptome data available on the database in October 2017). When single haploid male of each *Varroa* species reads were mapped onto the Vdes_3.0 reference genome, a total of 1,404,212 quality-filtered SNPs were detected. Relative to Vdes_3.0 genome size, sequence divergence with Vjacob_1.0 amounted to 0.38% across the nuclear genome sequence.

**Identification and confirmation of *Varroa* mite species and mtDNA lineages.** Mapping and pairwise whole mitochondrial alignment (16,505 bp) showed that Vdes_3.0 diverged of only 0.1% to the reference *V. destructor* mitogenome (NC004454.2)[53,54]. Despite presenting similar morphological phenotypes mainly distinguishable through female body size[30,55] (Fig. 1), the *V. jacobsoni* mitochondrial genome diverged from Vdes_3.0 mtDNA at 5.1% of all nucleotides.

Given that *COX1* is a standard marker used to identify the *Varroa* mite haplogroup, this region was extracted and realigned with 26 described sequences from native and introduced populations. Divergence within the complete *COX1* sequences

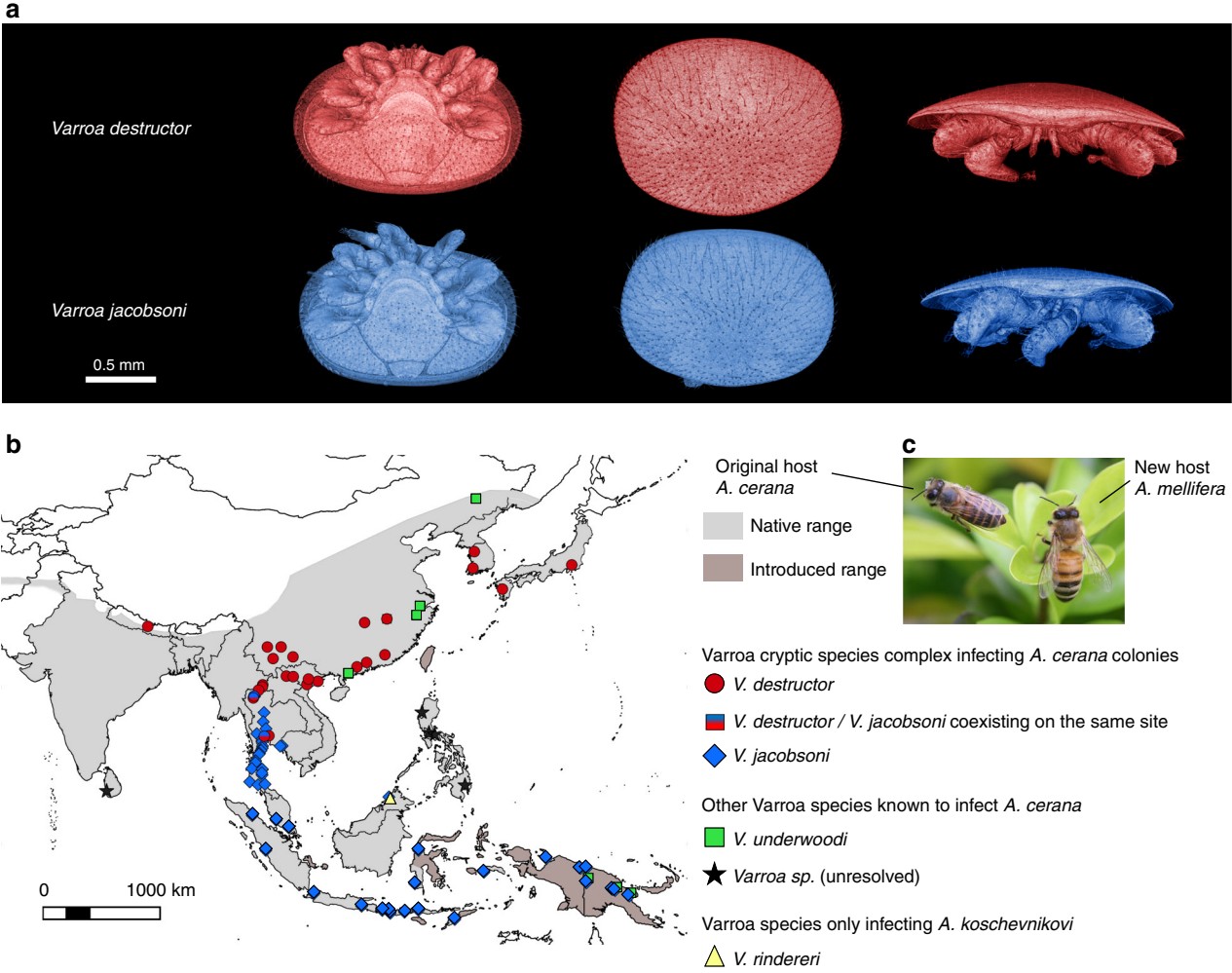

**Fig. 1** *V. destructor* and *V. jacobsoni* are morphologically similar sister species originally parasitizing the eastern honey bee (*A. cerana*). These species were recognized upon the basis of quantitative morphometric and genetic data[28]. Morphological differences are restricted to differences in body size and shape, as shown in **a** 3D surface models comparison of two fully sclerotized females. **b** Both mites parasitize *A. cerana* as their original host and can coexist in sympatry, occasionally even at the same apiary[47,71], but are mainly parapatric[121]. **c** With the introduction of *A. mellifera*, *V. destructor* can be found on this novel host throughout the original *V. jacobsoni* range. *V. jacobsoni* has also extended its range into Papua New Guinea on *A. cerana* followed by a shift to *A. mellifera*, where *V. destructor* is currently absent[31]

of the *V. destructor* and *V. jacobsoni* reference genomes rose up to 5.9%. The partial *COX1* sequence (426 bp) of Vdes_3.0 clustered with *V. destructor* sequences and was clearly distinct from Vjacob_1.0, which clustered with *V. jacobsoni* haplotypes (Fig. 2a, b). The concatenated *COX1*, *ATP6*, *COX3*, and *CYTB* sequence (2696 bp) from the Vdes_3.0 mitochondrial genome was identical to that of the invasive Korean K1-1/K1-2 sequence (Fig. 2c).

By contrast, the *V. jacobsoni* (i) partial *COX1* region (426 bp) was a unique new haplotype with 99% nucleotide similarity to Java (one transition point mutation G/A) and Bali (two transitions) haplotypes, and (ii) the concatenate sequence (only *COX1*, *ATP6*, and *COX3*) was also a distinct haplotype from the only referenced *V. jacobsoni* Laos 1 and 2. We named Vjacob_1.0 mtDNA haplotype as Papua New Guinea, which has been formerly reported in host switch events on *A. mellifera*.

**Orthologue genes between *Varroa* and other Parasitiformes.** The arthropod genome database at NCBI presently contains only two other published representative Mesostigmata genomes: the brood parasitic honey bee mite *Tropilaelaps mercedesae* (GCA_002081605.1)[56] and the non-parasitic Western predatory

mite *Metaseiulus* (=*Typhlodromus* or *Galendromus*) *occidentalis* (GCA_000255335.1)[57]. Orthologue prediction was performed by comparing annotated *Varroa* genes to those of the two other Mesostigmata species gene sets and those of two additional Acari: the parasitic tick *Ixodes scapularis* (GCA_000208615.1) and the free-living two-spotted mite *Tetranychus urticae* (GCA_000239435.1). The tick presents the largest set of genes ($n = 20,486$) and also possessing a haploid genome more than four times larger 1.76 Gb[58]. Orthology prediction analysis using the six genomes clustered the genes by domain into 7502 super-orthologue groups consisting of orthologues and inparalogues[59]. A total of 11,123 orthologue groups were predicted and 42.8% were shared by the five parasitiformes and the outgroup spider mite *Tetranychus* (Supplementary Data 1). The concatenated phylogenetic tree built on the genomic sets based on 4758 genes confirmed the close relationship between the three mite species compared to other Acari (Fig. 3). The honey bee ectoparasitic mites, *Varroa*, and *Tropilaelaps* shared 813 lineage-specific orthogroups (akin to gene families) of which 65.9% were Varroidae-lineage-specific. The two sibling *Varroa* species shared 9628 orthologous genes including 536, which were private to Varroidae.

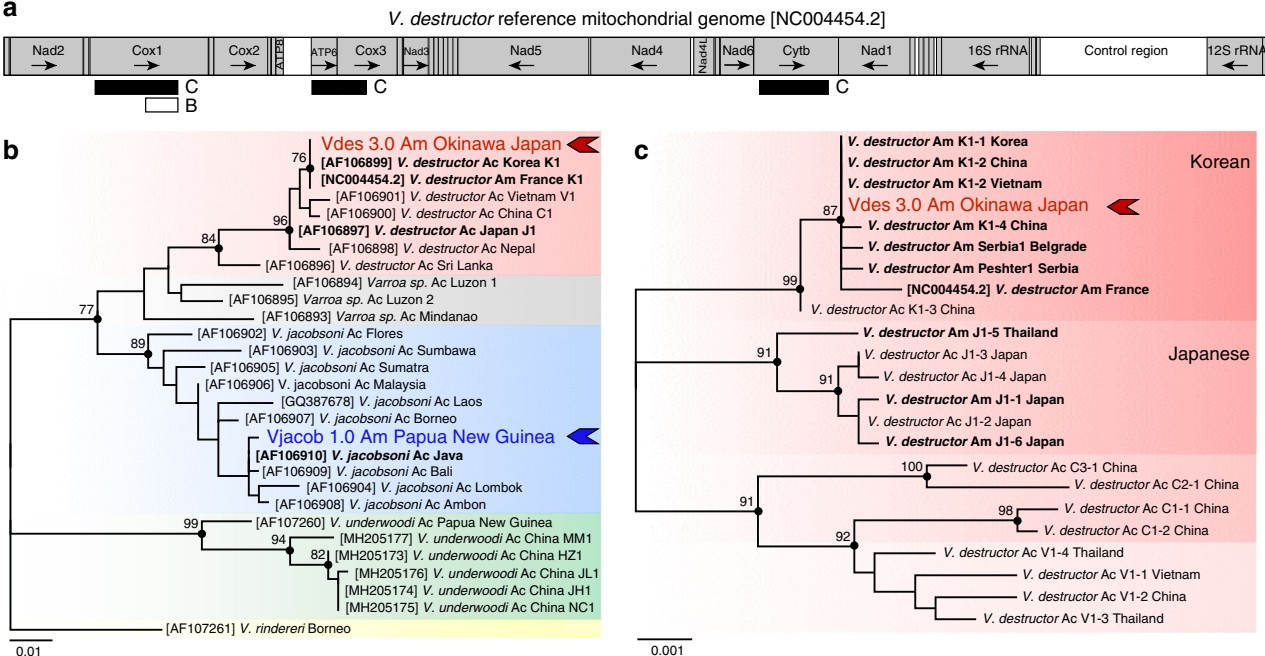

**Fig. 2** Inter and intraspecies mtDNA variability across the genus *Varroa* and host-switched lineages. Different mitochondrial markers used in the literature to discriminate the four *Varroa* species (and unresolved one) based on partial *COX1*[30] (**a**, **b**) and intraspecies lineages using larger *COX1, ATP6, COX3* and *CYTB* regions[69,78] (**a**, **c**). The two cryptic species *V. destructor* (red) and *V. jacobsoni* (blue) are genetically divergent as shown by the unrooted phylogenetic tree of *Varroa* mite partial *COX1* sequences (**b**). Variability of *V. jacobsoni* is higher than that of any other *Varroa* species and the reference genome (blue arrow) corresponded to one of the previously detected haplotypes switching on *A. mellifera* in Papua New Guinea[31] (**b**). Several haplotypes from the Korean and Japanese lineages successfully jumped on *A. mellifera* (bold)[69,78] but only K1-1/K1-2 is quasi-cosmopolitan and can even be retrieved in the native range of the Japanese lineage as illustrated by its presence in Okinawa (red arrow) (**c**). Ac identified from *Apis cerana*, Am identified from *Apis mellifera*

Orthologous genes specific to the Parasitiformes ancestral branch (i.e., before *T. urticae* and other Acari species split) was enriched for molecular functions and biological processes involved in several transporters and channel activities (Supplementary Data 2). When considering the Mesostigmata ancestral branch (i.e., before *M. occidentalis* and the three honey bee parasites species split), GO terms related with structural constituents of the cuticle were the most represented. At the Varroidae ancestral branch level (i.e., before the two species split), orthologous genes showed GO terms mainly related to endonuclease activity.

**Divergent patterns of selection among honey bee mites.** A total of 234 and 225 genes were detected under positive selection for *V. destructor* and *V. jacobsoni* branches, respectively, though only 13 were shared among this list (2.8%) (Fisher's exact test $p = 0.003$) (Fig. 3). Additionally, 40 genes were detected under positive selection when looking at the Varroa ancestral branch, i.e., before the two species split. On these set combined, *V. destructor* and *V. jacobsoni* presented 81.0% and 80.4% species-specific genes with positive selection signals, respectively (Supplementary Fig. 2 and Supplementary Data 3). The remaining proportion of positively selected genes shared involved the other honey bee mite *T. mercedesae* rather than with any other Acari. The genomic regions associated with genes under positive selection were found and distributed along all major chromosomes of *V. destructor*, and, assuming preservation of synteny, in *V. jacobsoni* as well (Fig. 4a).

Semantic distribution of the GO terms for the genes under positive selection in each *Varroa* species showed non-significant overlap ($p = 0.385 > 0.050$). The few terms that overlapped involved biological processes in essential functions such as the molting cycle—chitin-based, fatty acid, and lipid metabolism and

responses to pH (Fig. 4b). More specifically, chitin is the main constituent of the mite cuticle hardening it against both bees and the environment. Its metabolism varies throughout the mite life cycle, especially during reproduction or molting[42]. Fatty acid metabolism could be related to the feeding on the fat body on the host, as well as oogenesis. Although few overlaps exists when GO terms list were reduced using Resnik's measure, the REVIGO scatter plot and treemap visually suggest functionally different routes to host adaptation (Fig. 4b, Supplementary Data 4). In *V. destructor*, GO terms were mainly associated with (1) regulation of membrane depolarization, (2) retinal cone cell development and myofibroblast differentiation, (3) mRNA cis splicing via spliceosome whereas GO terms in *V. jacobsoni* related to (1) protein processing involved in targeting to mitochondria, (2) vitamin K metabolic process, (3) response to pH and other processes detailed in (Supplementary Fig. 3).

Tandem duplication was detected in 51 genes for *V. destructor*, 35 in *V. jacobsoni* and 70 in *T. mercedesae* (Fig. 3). Numbers of duplication events and gene clusters within a species varied from two to six units and generally involved adjacent chromosomal locations in *V. destructor* (Fig. 5a, Supplementary Data 5). In duplicated genes found only in *V. jacobsoni*, 23 had no orthologs in the *V. destructor* genome. However, most duplications are on the same scaffold with the exception of the ABC gene family where genes are present on two scaffolds. Manual checks of duplicated clusters with more than two genes validated all of the duplications except for the Histone 2 A family where *LOC111255102* gene located on the minor scaffold BEIS01000444.1 was removed since it is 100% identical to *LOC111250948* located on chromosome 1 and may be an artefact of mis-assembly.

Only five duplicated gene clusters were shared by all three honey bee parasites. These amounted to 9.8% and 14.3% of

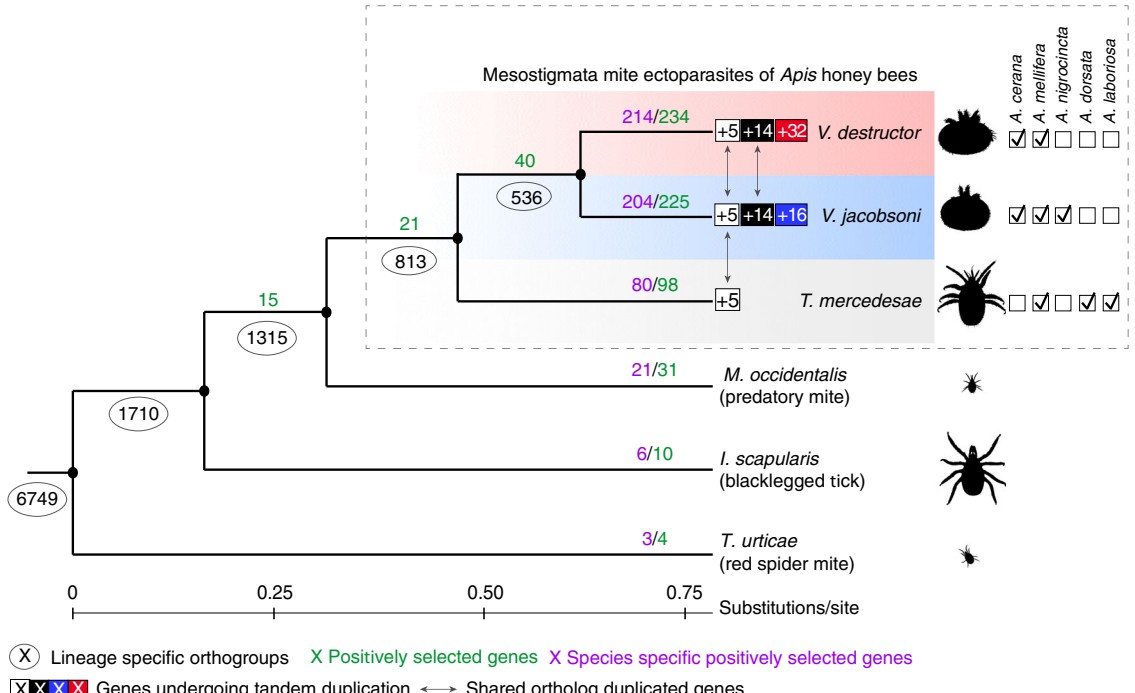

**Fig. 3** Positive selection and gene duplication of orthologous genes in parasitic and free-living Acari. For the honey bee mite parasites, host ranges are shown as checkboxes and individual records had to meet the following conditions: (1) Mites had to be found reproducing or observed in bee colonies in independent surveys, and (2) the identity of cryptic *Tropilaelaps* and *Varroa* species was confirmed by molecular markers (e.g., *COX1* barcoding[30,122]). The number of orthologous genes in each lineage is circled at the nodes. The number of positively selected genes for each branch is shown in green, with the number of species-specific genes shown in purple. Gene duplications within honey bee parasitic mites are shown in boxes: white squares = shared genes, black squares = *Varroa*-specific genes by, red and blue squares = *V. destructor* and *V. jacobsoni* specific genes, respectively. The two *Varroa* mites have similar host ranges, through *V. jacobsoni* also occurs on *A. nigrocincta*, a close relative of *A. cerana*, possibly as a result of a recent secondary host shift[30,55]. Despite sharing the same ancestral host, the *Varroa* sister species show different patterns of adaptive molecular evolution and exhibit different gene expansion patterns, suggesting different evolutionary trajectories

detected tandem gene duplications for *V. destructor* and *V. jacobsoni*, respectively (Fig. 2, Supplementary Data 6). GO terms associated with these duplicated gene clusters related to chitin, amino sugars, drug and carbohydrate derivatives. Conversely, the two *Varroa* sister species shared 14 gene clusters undergoing tandem duplication, representing 27.5% and 40.0% of the total size for *V. destructor* and *V. jacobsoni*, respectively. When summarizing the list of duplicated shared genes by both *Varroa* species, GO terms were associated with biological processes in glycerophospholipid catabolism, body development, regulation of DNA-templated transcription/elongation and amino-sugar metabolism (Supplementary Data 7). Finally, 62.7% of the duplicated genes were exclusive to *V. destructor* whereas 45.7% were only detected in *V. jacobsoni*. Genes duplicated in *V. destructor* were associated with GO terms related to biological processes such as mechanical/external stimulus, regulation of endoribonuclease activity and phagocytosis (Fig. 5b, Supplementary Data 7-8). In contrast, genes duplicated in *V. jacobsoni* were involved in biological processes of the skeletal myofibril assembly, Golgi calcium ion transport, and striated muscle contraction (Fig. 5b).

**Chemosensory and cytochrome P450 genes in honey bee mites.** *Lipid carrier proteins:* Based on the comparison of six amino acid sequences of OBPs previously described from *V. destructor* legs transcripts[60], we found seven homologs (with conserved domain IPR036728 or OBP-like) in *V. destructor* annotated genome (Supplementary Fig. 4). Each OBP gene found in *V. destructor* presented an ortholog in *V. jacobsoni* coding for an identical OBP sequence (Supplementary Data 9). An analogous pattern of strong conservation was found for eight NCP2s (IPR003172) or

similar with a Ganglioside GM2 activator conserved domain (IPR028996) in both sibling *Varroa* mites (Supplementary Fig. 5). None of the genes encoding for OBP and NCP2 showed signatures of positive selection or tandem duplication in *Varroa* sister species.

*Membrane-bound receptors:* A total of four genes per *Varroa* genome were found to encode for GR proteins presenting either the 7m_7 or Trehalose receptor conserved domains (IPR013604 and IPR009318, respectively). All four GR were closely related to GRs proteins found either in the honey bee parasitic mite *T. mercedesae* or in the parasitic tick *I. scapularis* (Supplementary Fig. 6). As previously reported, IGR protein profiles are the largest in *V. destructor*. According to the IGR phylogeny, this is also the case for *V. jacobsoni* with 35 identified genes (mainly presenting IPR001828, IPR019594 or IPR001320 signatures) (Fig. 6). Among these, 19 proteins found in *Varroa* genomes had closest homolog in *T. mercedesae* indicating high conservation of the gene family within honey bee parasites. Interestingly, three genes encoding for IGR or IGR-like proteins out of the 37 identified in *V. destructor* had no orthologous in *V. jacobsoni*. The opposite situation was also true but only for one gene in *V. jacobsoni* (*LOC111272277*). Additionally, divergent selective trajectories were found as a glutamate [NMDA] receptor subunit 1-like gene (*LOC111270396*) was under positive selection for *V. jacobsoni* while a metabotropic glutamate receptor 3-like gene (*LOC111245193*) show positive selection signal for *V. destructor*. Finally, despite a small number of SNMPs (with conserved domain IPR002159) detected in *V. jacobsoni*, we also found that one scavenger receptor class B member 1-like gene (*LOC111267160*) was under positive selection but none for *V. destructor* (Fig. S7).

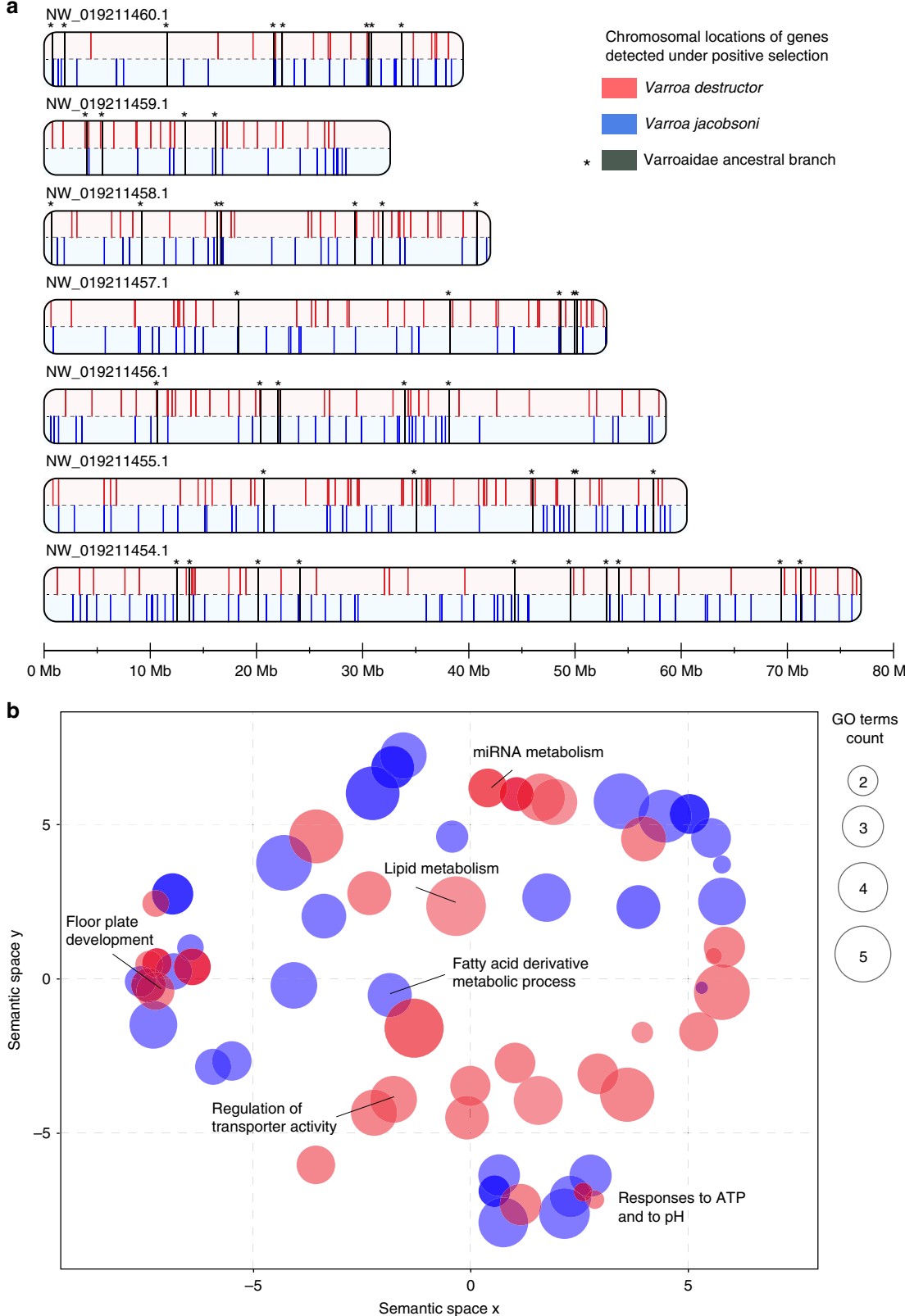

**Fig. 4** Genes and pathways under positive selection in the *Varroa* sister species. **a** Red and blue bars (in 5 kb windows) represent locations of genes in *V. destructor* and *V. jacobsoni*, respectively. *V. jacobsoni* data are mapped to *V. destructor* scaffold positions for comparison. Black asterisks indicated genes under positive selection, shared by both species (n = 12) and detected in the *Varroa* ancestral lineage prior to the split of the two species (n = 40). **b** Semantic space analysis of significantly enriched GO terms (BP, CC, and MF) over-represented in genes detected under positive selection for *Varroa* mites Bubble color indicates the species for which GO terms were enriched (all *p*-value < 0.05) and size indicates the frequency of the GO term found in the GOA database. There was little overlap in analyses at both gene (2.8%) and functional levels (0.8%), suggesting different selective pressures on the two sister species since they split

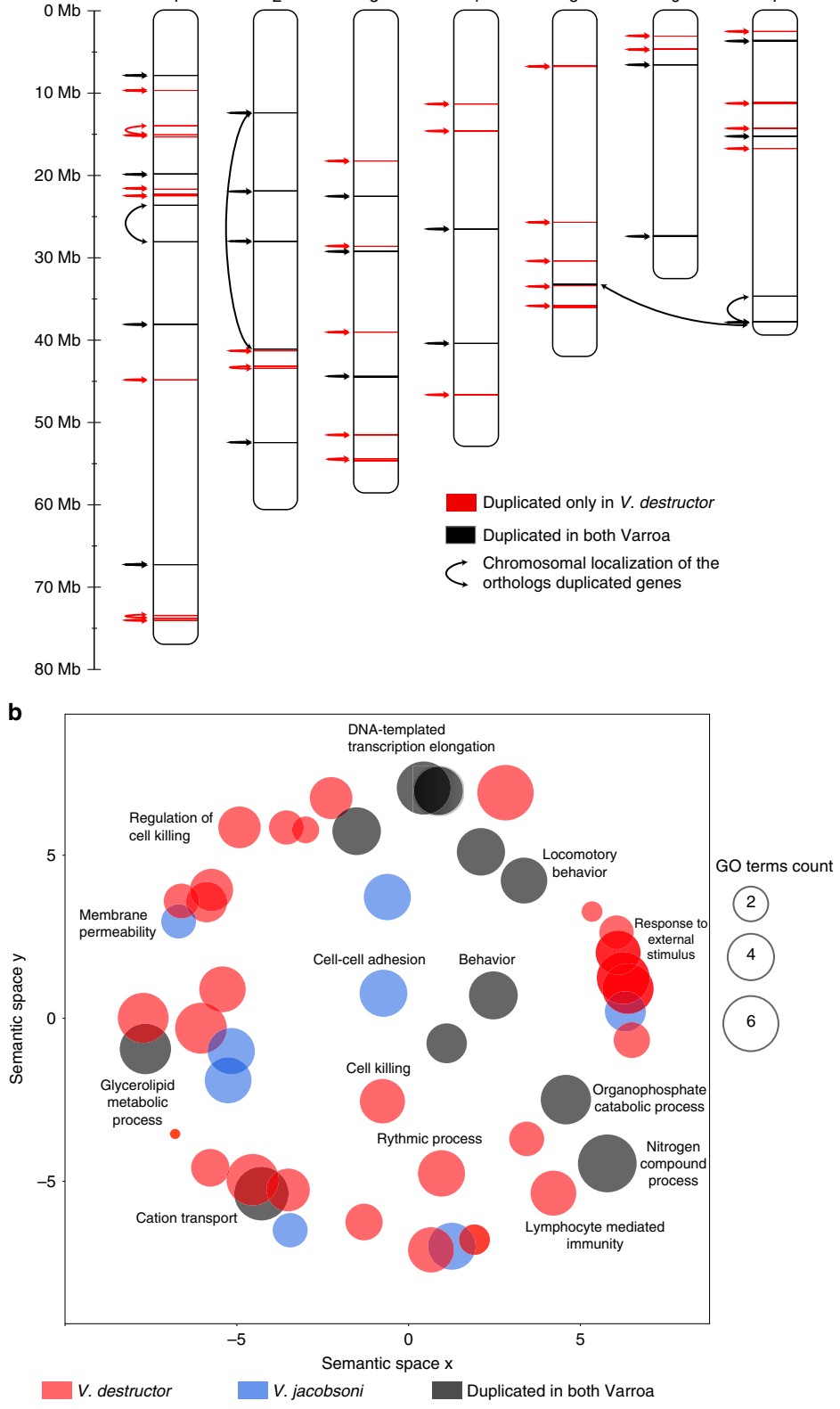

**Fig. 5** Duplicated genes in *V. destructor* are found throughout the genome and are involved in different biological pathways. **a** Chromosomal location and similarity of duplicated genes (arrows) for *V. destructor*. Different genes and chromosomal regions underwent duplication in the two species. **b** Cluster analysis of significantly enriched GO terms for biological processes over-represented among duplicating genes in *Varroa* mites. Not only were most of the GO terms species-specific, but they also comprised non-overlapping categories of biological processes. Neither gene duplication nor selection analysis suggests a substantial degree of parallel evolution in these mites

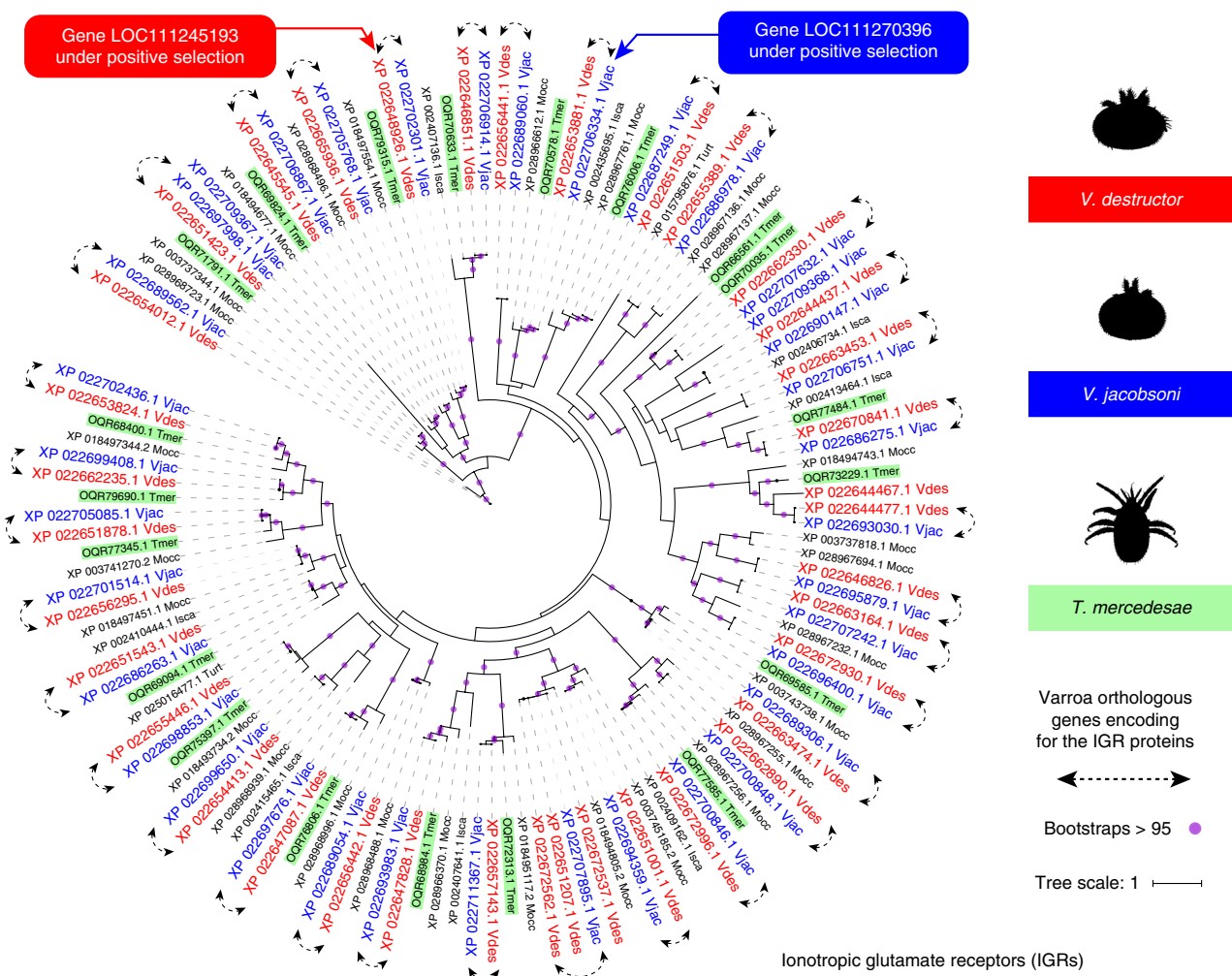

**Fig. 6** The largest IGR repertoire in honey bee parasites show divergent selective trajectories for host and environment chemosensing between *V. destructor* and *V. jacobsoni*. The phylogenetic tree was constructed using 153 amino acid sequences from Acari *V. destructor* (Vdes = genome and Var = protein sequences[83]), *V. jacobsoni* (Vjac), *T. mercedesae* (Tmer), *M. occidentalis* (Mocc), *I. scapularis* (Isca), and *T. urticae* (Turt) which were aligned with MAFFT (see list in Supplementary Data 9). Best fit model computed for the tree is VT + R5 using IQ-TREE. Bootstrap values were estimated using an SH-like aLRT with 1000 and bootstraps over 95% are shown by a purple circle

*Ecdysteroids and Halloween genes:* All CYP enzymes sequences from *V. destructor* and *V. jacobsoni* annotation were extracted by filtering gene description list containing cytochrome p450 or exhibiting the associated conserved domain cl12078 or pfam00067. A total of 36 and 38 CYP (one isoform per gene) were aligned together with the seven Halloween genes reported from *D. melanogaster* and *A. aegypti* and other CYPs from Acari. Previously, only some members of this pathway, namely orthologues of *spook*, *shade*, and *disembodied* were found from a genomic sequence generated by 454 pyrosequencing[45]. Homologs proteins of CYP302A1 and CYP314A1 (~32% and ~28% sequence identity to *disembodied* and *shade* of *D. melanogaster*, respectively) were found for both Varroa sibling species but surprisingly no homolog *spook* (CYP307A1) was found for *V. jacobsoni*, though there was one in *V. destructor* (Fig. 7). Phylogenetic relationships among Varroa mites, *T. urticae* and *M. occidentalis* and *D. melanogaster* CYP315A1 sequences (~36% identity with the fruit fly) suggest that both *V. destructor* and *V. jacobsoni* Varroa mites possess *shadow* homologs. Likely, *shroud* and *phantom* are also presents in both Varroa species but not *neverland*. In the CYP family, we detected three genes under positive selection only in *V. jacobsoni*, including *shadow*-like

(*LOC111271433*) and *disembodied* (*LOC111265704*) (Fig. 7 and Supplementary Data 9).

*Other genes possibly related to nutrition and detoxification:* Some annotated genes detected under positive selection or undergoing tandem duplication in *Varroa* have been linked to play a role in external stress tolerance (e.g., in-hive temperature), nutrition, molting, reproduction, and metabolizing toxic xenobiotics possibly leading to acaricide resistance (detailed in Table 2). On *V. destructor* chromosome 4 (NW_019211457.1), two duplicated genes were also under positive selection and were involved in cuticular proteins pathway. These genes are involved in physical properties of the mite cuticle, which in addition to providing protection may constitute the first barrier to reduce the penetration of external chemical agents[61]. On the other hand, two duplicate clusters were also detected undergoing positive selection and annotated as part of the ABC family (ATP-Binding Cassette sub-family A) and to aminopeptidase M1-B-like, in *V. jacobsoni*. Finally, the most remarkable tandem duplication occurred on Vdes_3.0 chromosome 1 (NW_019211454.1) with six duplicated genes coding for heat shock 70 kDa protein. These were not detected for *V. jacobsoni* (Table 2). Both positive gene selection and duplication patterns in the *V. destructor* and *V. jacobsoni*

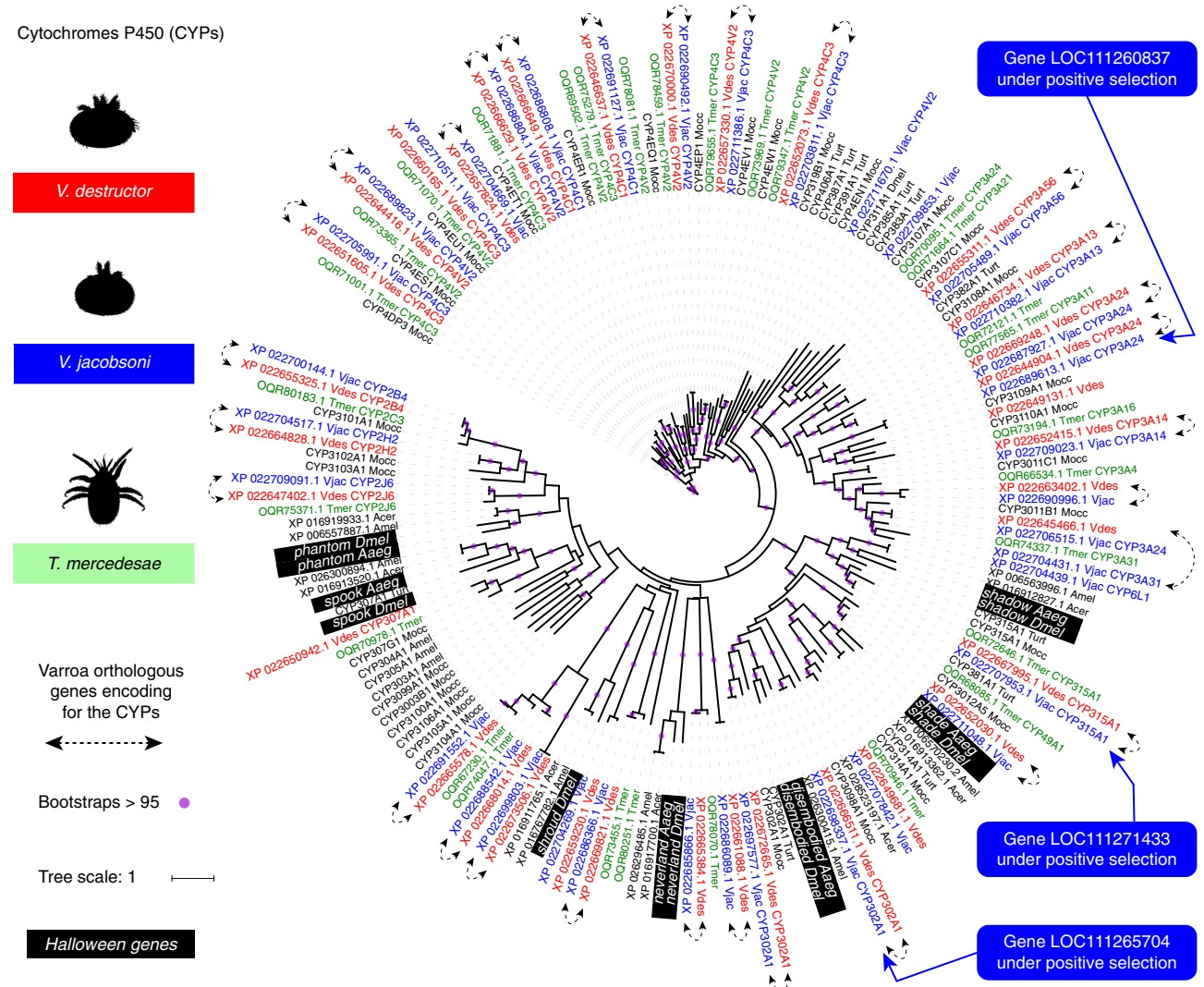

**Fig. 7** CYP repertoire in Varroa mites is broader than expected with two functionaly important Halloween genes *shade* and *disembodied* under positive selection in *V. jacobsoni*. The phylogenetic tree was constructed using 174 amino acid sequences from Acari *V. destructor* (Vdes = genome and Var = protein sequences[83]), *V. jacobsoni* (Vjac), *T. mercedesae* (Tmer), *M. occidentalis* (Mocc), *I. scapularis* (Isca), and *T. urticae* (Turt). Additionally, sequences from the fruit fly *Drosophila melanogaster* (Dmel), mosquito *Aedes aegypti* (Aaeg) and the honey bee *A. mellifera* (Amel) and *A. cerana* (Acer) were downloaded form NCBI and aligned with MAFFT. Best fit model computed for the tree is LG + R5 using IQ-TREE. Bootstrap values were estimated using a SH-like aLRT with 1000 and bootstraps over 95% are shown by a purple circle. *V. destructor* was previously believed to have only *disembodied* (CYP302a1), *shade* (CYP314a1), and *spook* (CYP307a1) homologs of the seven Halloween genes (background in black)[45]

genomes and related GO terms showed that the two sister species have undergone dissimilar evolutionary trajectories.

## Discussion

In this study, we developed high-quality de novo reference genomes and annotations for economically important honey bee parasites *V. destructor* and *V. jacobsoni*. Because *Varroa* mites are, to a large extent, responsible for the global honey bee crisis, and research efforts worldwide target these parasites, these genomic resources respond to a pressing need[37]. Improved genome assembly and annotation with transcriptomic data from different life stages produced the most comprehensive gene catalogue to date, as reflected by our discovery of previously missing Halloween genes homologs[45]. The supposed absence of such genes has led to speculation on control measures development or that host counter adaptation could in the future target this pathway[46]. Our analysis also revealed that the two *Varroa* species

share 99.7% sequence identity across the nuclear genome. Yet, after the two species have split, they have gone on largely dissimilar evolutionary trajectories, as evidenced by different patterns of positive selection and gene duplication. In particular, signals of positive selection were found in functionally important genes involved in chemosensing, molting, and reproduction. Thus, it seems likely that the two species underwent different patterns of adaptation while continuing to exploit the same host, the eastern honey bee (*A. cerana*). Differences between the two species most likely result from local adaptation to host geographic subspecies, the environment or their interaction. Though not enough data exist to distinguish between these alternatives, below we summarize what is already known and outline promising avenues for investigation.

Inferring what constitutes the ancestral ecological niche of the two mites is key to understanding what selective pressures they have faced and how they responded to them. *V. destructor* and *V. jacobsoni* are very similar morphologically, and until 20 years ago

**Table 2 Genes undergoing positive selection or duplication, that have been implicated in tolerance to external stressors and stimuli**

| | V. destructor (Vdes_3.0) | | V. jacobsoni (Vjacob_1.0) | |
| --- | --- | --- | --- | --- |
| | Annotated gene under positive selection | Annotated gene undergoing duplication | Annotated gene under positive selection | Annotated gene undergoing duplication |
| **Stress from in-hive temperature** | | | | |
| Heat Shock 70 Kda protein-like[123] | | 6 | | |
| **Nutrition** | | | | |
| Guanine metabolism[124] | | | 1 | |
| **Detoxification** | | | | |
| Glutathione-S-transferase[80] | | | 1 | |
| Sodium channel related[125] | 4 | | 2 | |
| Transmembrane proteins | 3 | | 4 | 2 |
| G-protein coupled receptors | 1 | | | |
| Esterases[126] | 2 | | | |
| GABA-gated ion channels[127] | 1 | | | |
| ATP-binding cassette gene family[128] | 1 | | 2 | 2 |
| **Molting and reproduction** | | | | |
| Ecdysteroids pathway with cytochrome P450[45] | | | 3 | |
| Cuticular protein | 2 | 4 | | 1 |
| **Nutrition and reproduction** | | | | |
| Fatty acid metabolism[118,129,130] | 3 | | | |
| **Chemosensory receptors** | | | | |
| Sensory neuron membrane protein | | | 1 | |
| Ionotropic glutamate receptor | 1 | | 1 | |

Research on Acari and *V. destructor*, in particular, has identified a large number of genes involved in stress and detoxification (possibly leading to acaricide resistance), nutrition, reproduction and chemosensing in interaction with the host. Intriguingly, many of these genes appear to be under ancestral positive selection, as *V. jacobsoni* has not been extensively targeted by acaricides on *A. mellifera*, and most of the evolutionary history captured by positive selection analyses occurred in *A. cerana*, given that coevolution with *A. mellifera* is a relatively recent evolutionary phenomenon, particularly for *V. jacobsoni*.

they were mistakenly considered the same species[30]. Their host specificity overlaps as they both parasitize the eastern honey bee *A. cerana* throughout Asia. The only exception reported for the Indonesian *A. nigrocincta* infested by *V. jacobsoni*, though possibly as a result of a recent host switch from *A. cerana*[30,62]. The most recent common ancestor of *V. destructor* and *V. jacobsoni* might have also been a parasite of the *A. cerana* lineage. This hypothesis is supported by the host range of *V. underwoodi*, another closely related mite that also parasitizes *A. cerana* in addition to *A. nigrocincta* and *A. nulensis*[63,64]. With no known varroid fossils, no information exists about speciation timing in *Varroa* mites. Based on the *COX1* gene, a recognized molecular clock for insects with a divergence rate of 2.3% per Myr[65], the divergence between *V. destructor* and *V. jacobsoni* of 5.9% suggests that the two species split ~2-3 Myr ago. This is after the split between *A. cerana* and *A. mellifera*, which occurred 6–9 Myr ago based on *COX1* and *ND2*[66,67] to 17–19 Myr using a combination of mitochondrial, ribosomal, and nuclear *loci*[68]. Thus, host ranges and molecular estimates suggest that most *Varroa* evolution occurred on ancestors of *A. cerana*.

The major ecological difference between these mites is their parapatric distribution (Fig. 1). Whether parapatry resulted from their respective biogeographic history, adaptation to different subspecies of *A. cerana*, or is shaped by competition is not known. In fact, even the extent of the geographical overlap between the mite distributions is uncertain, since few surveys actually used molecular markers for identification and most have focused on associations with *A. mellifera*[30,64,69,70]. Yet, parapatry is clearly not due to the inability of the mites to complete their phoretic life stage outside the hive, as host-switched *V. destructor* successfully expanded its range following the range extension of *A. mellifera*. While allopatry could result from adaptation to local

hosts/conditions or competitive exclusion, both mites have been reported from the same Thai apiary on *A. cerana*[47,71]. Unfortunately, there are not enough data on *A. cerana* infestation under natural conditions to determine the likelihood that the two *Varroa* species interact and compete. One survey in southern Thailand found 98% parasitism by *V. jacobsoni* under natural conditions, albeit at variable levels of infestation[72], suggesting that encounters between the two species are indeed likely. If the divergent evolutionary history of the two *Varroa* species resulted from competition, we predict that long-term coexistence within a hive of *A. cerana* should be unlikely, a hypothesis that we hope will be tested in the future in the zone of sympatry.

In addition, or as an alternative to interspecific competition, it is possible that the divergence between *V. destructor* and *V. jacobsoni* was driven by local adaptation to different *A. cerana* populations. For instance, *V. destructor* was found to infect the Northern, Himalayan and the Indo-Chinese *A. cerana* morpho-clusters identified[48] whereas *V. jacobsoni* was reported on the Indo-Chinese and the Indo-Malayan one. These *A. cerana* populations differ in their nesting behaviour, such as the number of combs or even in their body size[73]. In China alone, *A. cerana* populations have been shown to be as genetically divergent for more than 300,000–500,000 years as recognized subspecies of *A. mellifera*[74]. In addition to genetic differences between the hosts, the climactic environments across Asia are highly variable and could result in local adaptation, though, as mentioned earlier, this did not prevent range expansions after switching hosts (Fig. 1). Consequently, the dissimilar selective pressures observed in the mites may have resulted from selection due to different host genotypes.

Understanding the dynamics of co-infestation by both mites will become important if the host-switched *V. destructor* and *V.*

*jacobsoni* are ever found to co-occur on *A. mellifera*. Presently, host-switched *V. jacobsoni* is restricted to Papua New Guinea, where *V. destructor* is absent[31]. However, if its range expands, it may impose an additional parasitic burden on *A. mellifera*, but whether the effects of this extra parasite are cumulative or whether the parasites will simply replace each other locally is unknown. Our data suggest that if the parasites are able to tolerate each other within *A. mellifera* colonies, given their different suites of molecular adaptations, they may in fact impose a greater cost than either parasite alone. Clearly, this is another avenue where further experimental work is urgently needed.

Likewise, it is unknown whether chemical treatment options developed for *V. destructor* will also be effective against *V. jacobsoni*. Interestingly, both mite genomes show signs of selection on a range of genes previously found to be involved in stress and pesticide resistance (Table 2). Since *V. destructor* has evolved resistance to acaricides, it cannot be ruled out that positive selection resulted from recent acaricide pressures rather than coevolution with their host. By contrast, acaricides are not yet widely used on *V. jacobsoni*, though the number of genes under selection is similar, suggesting that most selection is ancient. In the case of *V. jacobsoni*, and possibly generally, these genes have the potential to become co-opted for pesticide resistance. This has been the case with pesticide tolerance in the sheep blowfly (*Lucilia cuprina*), where sequencing of museum specimens has shown that resistance variants pre-dated pesticide use[75]. Similarly, the two-spotted mite (*Tetranychus urticae*) appears to have co-opted existing detoxification machinery in a selective response to miticides[76]. However, more detailed population-level work comparing switched and unswitched populations of mites will be necessary to test this hypothesis. Museum specimens or other historical material would be particularly useful in understanding evolutionary trends[77].

Despite the economic importance of brood parasitic mites in *A. mellifera*, it is clear that various pressing research needs still exist concerning their biology, cryptic diversity in the native range, possible geographic ranges of host-switched species, and how co-infestations may proceed. These new reference genomes will allow researchers to pursue such investigations, develop better tools for *Varroa* control, and to better understand the invasion and evolution of *Varroa* on *A. mellifera*. In particular, we hope that these new resources will greatly benefit future studies to track the population genetic changes and demography of *V. destructor*'s worldwide spread, which has been difficult due to the low polymorphism of previously available markers[70]. More *Varroa* spp. are continuously testing new host[78,79] and may remain undetected. Finally, the recent boom in -omics methods for the study of honey bee parasites and diseases underlines the need to improve our knowledge of genome structure and organization to investigate the genetic basis of newly reported acaricide resistance, virulence level or even new host tolerance. Such knowledge combined with progress in novel manipulation tools, such as RNAi[80,81], and more experimental tools like gene drive[82], may provide ways of specifically controlling mite pests without harming honey bees or other insects.

## Methods

**Sequencing strategy and *Varroa* mite collection**. Haploid males (~0.7 mm by 0.7 mm) are only found inside capped brood cells after initiation of reproduction while diploid females (~1 mm by 1.5 mm) can be found in the brood and on adult honey bees. All *V. destructor* samples were collected from *A. mellifera* colonies of Okinawa Institute of Science and Technology (OIST) experimental apiary in Japan (26.477 N 127.835 E) in August–September 2016. One mature male *V. destructor* was collected within a worker cell at the red-eye pupa developmental stage (day 15–16) using a brush and kept in absolute ethanol at −80 °C. Mature adult females were collected from adult honey bee workers. In order to obtain a large number of mites infecting adult workers, we modified the standard powdered sugar method[55]

for an entire managed colony as follows: one hive box containing three to four frames with honey bee workers and no queen was placed on top of a collecting white tray; approximately 500 g of icing sugar was then applied using sieve filter powder strainer; the sugar and honey bees were shaken in the tray for 2 min and then separated using a sieve filter, placing the sugared-workers directly within the hive. This process was repeated on adjacent colonies. *Varroa* mites were separated from their hosts and trapped in the icing sugar. Following a water rinse on a gauze mesh, a total of 1207 alive *Varroa* females were snap frozen at −80 °C until laboratory processing (inactive, sluggish or dead mites were discarded). Samples of *V. jacobsoni* were obtained from its first detection survey where it was found reproducing in the western honey bee[31]. One mature male (VJ856) collected from *A. mellifera* drone brood cell in Papua New Guinea (between EHP and Henganofi border, 30/05/2008) was used to prepare a whole-genome library.

One sclerotized mature female *V. destructor* (VDOKI-01) collected from OIST apiary in the reproductive phase and one sclerotized mature female *V. jacobsoni* (VJ347-11) collected from *A. cerana* originating in Java (Cililin central Java, 13/06/1998 by D. Anderson) were processed for X-ray microtomography (micro-CT) using a Zeiss Xradia 510 Versa scanner. 3D surface models were reconstructed from micro-CT outputs using Amira software to visualize the slight morphological differences when focusing on idiosoma ventral and dorsal views and whole body in profile (Fig. 1a).

**DNA extraction, genome sequencing, assembly, and annotation**. Genomic DNA was extracted from each single haploid male *V. jacobsoni* and *V. destructor* using QIAamp DNA Micro Kit (Qiagen) following manufacturer's instructions. The total amount of dsDNA was measured using Quant-iT™ PicoGreen™ dsDNA Assay Kit (Invitrogen). Short-insert of 250 bp paired-end libraries were prepared for both individuals using NEBNext® Ultra™ II DNA Library Prep Kit for Illumina® (Biolabs) and 10 PCR cycles. Size-selection and cleanup was applied using SPRI beads and subsequently checked using a Bioanalyzer High Sensitivity DNA kit (Agilent). Both paired-end libraries were each sequenced on two lanes of Illumina HiSeq 2500 at the OIST sequencing center in paired-end 250 cycle mode.

Prior to assembly paired-end reads were de-duplicated using ParDRe[83], and host and microbial contamination was filtered using DeConSeq (parameters: -i 97 -c 95)[84], screening against genomes of *Apis cerana*, *Apis mellifera*, as well as bacterial genomes and genomes of bee diseases and parasites. The cleaned reads were then merged using PEAR (parameters:–min-overlap 10–memory 80 G–threads 10 -n 200 -m 600 -p 0.0001)[85] and assembled using Newbler (v. 2.6) (parameters: -large -m -cpu 24 -mi 98 -siom 390 -l 1000 -a 500 -urt -novs -a 1000 -minlen 45 -ml 100)[86]. The contigs were then scaffolded using publicly available RNA-seq data assembled using Trinity[87,88]. For *V. destructor* further scaffolding was carried out using Hi-C, performed by Phase Genomics. Hi-C read pairs were aligned to the initial assembly using BWA-MEM with the -5 option[89], keeping only primary properly paired alignments (samtools with the -F 2316 filter[90]). Scaffolding was then performed using Phase Genomics' Proximo Hi-C scaffolding pipeline[91], which builds on the LACHESIS method[92].

To estimate genome characteristics, K-mers from cleaned reads were counted using jellyfish version 2.2.7[93] for the following values of K: 19, 25, 31, and 42 and coverage cutoff of 1000. Frequencies were only computed on HiSeq 2500 reads from one individual haploid male of each species. Computed and selected K-mer histograms for k = 42 were analyzed with GenomeScope to estimate genome size, kcoverage, number of duplicates[94]. We used the bioinformatically derived genome sizes for genome coverage estimation.

Small batches of adult females *V. destructor* (1,207 mites total) were pooled and crushed by pestle in 1.5 mL Eppendorf tubes. The powder was then lysed using proteinase K and lysis buffer at 56 °C for 20 min. Sample lysate was thus transferred to a Maxwell Cartridge. The remaining step was fully performed on the Maxwell 16 automated purification system (Promega) with the LEV Blood DNA mode. Library preparation for PacBio RSII sequencing was performed using SMRTbell Template Prep Kit 1.0 (Pacific Biosciences) following the manufacturer's recommendations. SMRTbell libraries were prepared and sequenced at the OIST sequencing center on 48 SMRT cells. These reads were trimmed using proovread[95] and then used to gap-fill the Hi-C scaffolded *V. destructor* assembly using PBJelly (parameters: -minMatch 8 -minPctIdentity 70 -bestn 1 -nCandidates 20 -maxScore -500 -nproc 24 -noSplitSubreads) (v. 15.8.24)[96].

Annotation for both genomes Vdes_3.0 and Vjacob_1.0 was carried out by NCBI using NCBI's automated genome annotation pipeline. This pipeline takes advantage of species-specific RNA-seq data, as well as extensive protein homology data stored in GenBank[52].

**Identification of *Varroa* mite strains using mtDNA sequences**. Most previous taxonomic work relied on mitochondrial fragments. We used mitochondrial analysis to determine and compare the lineages of *Varroa* species used for genome sequencing to the previously described haplotypes. The whole mitogenome of *V. destructor* (accession NC_004454.2[53,54]) was available from the NCBI database and used as a reference for read mapping. Raw Illumina® reads for both *V. destructor* and *V. jacobsoni* males were mapped onto the entire reference sequence (16,476 bp) using NextGenMap 0.5.0[97]. Mapped reads were sorted and PCR duplicates were removed using Samtools 1.3.1[90]. Maximum coverage of 500x was subsampled on all mapped reads using VariantBam 1.4.3[98]. Subsequently, variant calling was

performed using FreeBayes v1.1.0-46[99], with the following parameters others than the default: minimum mapping quality of 20 and minimum base quality of 5. Consensus mitochondrial sequence was generated for each individual using vcf2fasta from Vcflib library[100]. To check the divergence level across the nuclear genome, we also mapped haploid male data from each species to the Vdes_3.0 genome using the same mapping and variant calling parameters.

One standard method to identify Varroa species and haplogroup is to sequence a partial region of the COX1 gene (426 bp) from the mitochondrial DNA[55]. The same region was extracted from both Vdes_3.0 and Vjacob_1.0 generated mitochondrial sequence and aligned together with 26 Varroa spp. haplotypes previously described[30,69,79,101]. Given its worldwide bee pest status, V. destructor haplogroups were better described than V. jacobsoni and additional larger COX1, COX3, ATP6, and CYTB mitochondrial sequences were available on NCBI. Therefore, a more resolutive comparison within V. destructor was done by aligning concatenated sequences of the four genes with 22 haplotypes described previously[69,78]. Sequences were aligned using ClustalW, checked by hand and a consensus maximum likelihood tree was constructed after 5000 bootstraps in Mega 7.0.18[102].

A distribution map of the Varroa mites infecting A. cerana was built from reviewing literature (Fig. 1b). Conservatively, only the presence of Varroa spp. confirmed by mtDNA barcoding after the taxonomic revision was plotted[30,47,69,79,101,103–105] onto the potential native and introduced range of the Asian honey bee host[48,73] (Fig. 1c). The map was produced using QGIS 2.16.1 (Open Source Geospatial Foundation Project qgis.osgeo.org) and the public domain natural 10 m earth raster NE1_HR_LC github.com/nvkelso/natural-earth-raster). An additional interactive map showing the reported distribution of the 26 mtDNA haplotypes is available at github.com/MikheyevLab/varroa-denovo-genomes with layers control options.

**Gene orthology, selection, and duplication in Varroa mites**. The Orthonome pipeline (www.orthonome.com)[59] was applied to predict orthologues (pairwise and 1:1 orthogroups), inparalogues and lineage-specific gene expansions from the two parasitic Varroa mites' genomes sequenced and annotated here and four Acari annotated reference genomes available from NCBI (www.ncbi.nlm.nih.gov/genome/browse/#!/overview/Acari). The Orthonome pipeline uses phylogenetic information and microsynteny to identify gene copy number changes and tandem duplications between pairs of species using the MSOAR2.0 pipeline[106]. We included four outgroups which had fully sequenced genomes, including the eastern honey bee ectoparasitic mite Tropilaelaps mercedesae[56], the parasitic tick Ixodes scapularis[58]; the free-living Western predatory mite Metaseiulus occidentalis[107] and the free-living two-spotted mites Tetranychus urticae[108]. Total and pairwise/lineage intersection of orthologs gene sets were represented using Upset plots[109] using R. The 1:1 orthogroups with both Varroa species represented and >3 species were used for positive selection hypothesis tested on phylogeny using HyPhy to calculate the mean dN/dS and test using aBSREL (adaptive Branch-Site Random Effects Likelihood)[110]. This test allowed estimation of selection along specific branches of the tree, including ancestral branches. Only branches with a significant p-value (<0.05) after FDR corrections were considered as positively selected in each orthogroup. Additionally, tandem duplications in the three parasitic mites were identified from inparalogues in immediate vicinity of the orthologues. All orthologous genes in Varroa mites under positive selection and/or duplicated in tandem were checked by hand by using BLAST. Duplication events with more than two genes were systematically checked by aligning the annotated coding sequence (CDS) with ClustalW alignment implemented in Geneious R8 (Biomatters, Ltd., Auckland, New Zealand) and construct a tree with IQ-TREE[111].

Gene ontology (GO) enrichment analysis of the positively selected genes in honey bee parasitic mites were carried out using the GOstats R package[112]. The same process was applied for gene duplication in Varroa mites. Biological processes associated with these GO terms were summarized and visualized using REVIGO[113] (http://revigo.irb.hr). The semantic similarity measure using the Resnik's measure (SimRel)[114] and the threshold used was C = 0.5 (i.e., small). The results were then used to produce a scatter plot and treemap plot using the ggplot2 package. The same package was used to plot the localization of the genes under positive selection and/or duplicated onto V. destructor chromosomal scaffolds. Significance of overlaps between gene sets and associated GO terms were tested using Fisher's exact test with GeneOverlap R package. Code and raw plots for exploratory analysis of positively selected genes, duplication in tandem and GO term association are available online (https://github.com/MikheyevLab/varroa-denovo-genomes).

**Chemosensory and ecdysteroid pathway-related gene families**. Varroa mites are blind and rely on their chemosensory system to find a suitable host within a densely packed honey bee colony[40]. Chemical cues are transported to the nervous system by binding to soluble carrier proteins and membrane proteins in insects. Several homologous chemosensory protein families involved in these pathways have been identified in Acari and Varroa mites[41,43,44,60]. Here we used the available catalogue of protein sequences reported from V. destructor forelegs transcripts[44,60] containing six Odorant Binding Proteins (OBP), six Niemann-Pick disease protein, type C2 (NPC2), seven Sensory Neuron Membrane Proteins (SNMPs), three Glutamate Receptors (GRs), and 17 Ionotropic Glutamate Receptor (IGRs). We

identified these proteins in both V. destructor and V. jacobsoni annotated genomes using BLASTP and compiled predicted protein sequences (keeping one isoform if several exist for the same gene). We then aligned each family of protein sequences with other sequences from other Acari and insects using MAFFT version 7[115] (parameters: E-INS-i, the BLOSUM62 matrix, and MAFFT-homolog option with 100 homologs, E threshold = 0.1). We used IQ-TREE[116] to find the best substitution model for each family alignment, with state frequency estimated by maximum likelihood, 1000 ultra-fast bootstraps, 1000 replicate for SH-aLRT branch test. All generated phylogenetic trees were visualized and exported using iTOL v4[117].

We ran the same process for the cytochrome P450 enzyme (CYP) genes (including the Halloween genes), known to be involved in the steroid molting hormone ecdysone biosynthesis pathway in Arthropods. Ecdysteroids derivatives are also important as they are involved in the vitellogenin production and oogenesis initiation in V. destructor after feeding on its host[45,118]. To assess which CYP genes and in particular the Halloween genes were present in both V. destructor and V. jacobsoni genomes, we used sequences from the fruit fly Drosophila melanogaster, the mosquito Aedes aegypti, both honey bee hosts A. mellifera and A. cerana, as well as other Acari (same name code as in a previous study on M. occidentalis[119]).

**Statistics and reproducibility**. GO terms enrichment was conducted with the R package GOstats using the hypergeometric test for association of categories and genes. The test parameters for each species and each ontology (BP, CC, and MF) using gene ID from NCBI were as follow: p-value cutoff = 0.05, not conditional and with detection of over-represented GO terms (testDirection = over). Regarding the test conducted for the significance in gene overlap, a Fisher's exact two-tailed test with GeneOverlap R package and can be reproduced directly from the online supplementary data.

**Reporting summary**. Further information on research design is available in the Nature Research Reporting Summary linked to this article.

## Data availability

All data generated or analysed during this study are included in this published article (and its supplementary information files). V. destructor (Vdes_3.0) and V. jacobsoni (Vjacob_1.0) assembled reference genomes and annotation are available on NCBI database respectively with the accession number GCF_002443255 and GCF_002532875. These data are also available for search and annotation at the i5k Workspace@NAL (https://i5k.nal.usda.gov/)[130]. Raw Illumina and PacBio reads are also available in the Sequence Read Archive (SRA) under the Bioprojects PRJDB6279 and PRJNA391052. Raw data of both Varroa mites micro-CT scans are available upon request.

## Code availability

Online supplemental codes and map are available with an R markdown hosted from on github.com/MikheyevLab/varroa-denovo-genomes[120].

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

## Acknowledgements

We would like to acknowledge Yoann Portugal (Ecology and Evolution OIST) for helping with the fieldwork collection of *V. destructor* mites, Jo Tan (Ecology and Evolution OIST) for processing in laboratory *Varroa* males and Hiroki Goto (OIST DNA sequencing section – Onna, Okinawa) for leading and processing the PacBio RSII sequencing. We also thank Francisco Hita Garcia (Biodiversity and Biocomplexity OIST) for his support in preparation and scanning of specimens using the micro-CT. We thank Steven Aird (OIST technical editor) for carefully checking and helping to correct our manuscript. This work was supported by the Okinawa Institute of Science and Technology Graduate University, as well as grants from the Japan Society for the Promotion of Science (Kakenhi grants 16KK0175, 16H06209 and 18H02216) and the Australian Research Council (Future Fellowship Scheme FT160100178). Genome assembly was partially supported by Driscoll's, Inc.

## Author contributions

M.A.T. collected samples for sequencing, carried out analyses, interpreted the data, and wrote the manuscript. R.V.R. carried out analyses and wrote the manuscript. J.M.K.R. collected samples for sequencing. M.L.G. carried out the assembly and quality analyses. S.T.S. carried out the computational analyses for Hi-C genome assembly. I.L. carried out the experimental protocols for Hi-C genome assembly. A.K.C. conceived and designed experiments. J.D.E. conceived and designed experiments. A.S.M. conceived and designed experiments, carried out the assembly and analyses, interpreted the data, and wrote the manuscript.

## Competing interests

I.L. and S.T.S. declare the following competing interests: I.L. and S.T.S. are employees and shareholders of Phase Genomics, a company commercializing proximity-ligation technology. M.A.T., R.V.R., J.M.K.R., M.L.G., A.K.C., J.D.E., and A.S.M. declare no financial or non-financial competing interests.
