## [Peer Review File · Communications Biology]

Reviewers' comments:

Reviewer #1 (Remarks to the Author):

This study presents a revised genome for the ectoparasitic mite *Varroa destructor* along with new draft assemblies for the recently recognized distinct species *V. jacobsoni* and the more distantly related *Tropilaelaps mercedesae*. The new assembly for *V. destructor* has high contiguity and achieves nearly chromosomal-level assembly, although claims about it being the best assembled arthropod genome are unneeded, likely to be quickly superseded, and do not seem to consider the genomes of the insects *Drosophila* or *Tribolium*. The *V. jacobsoni* genome is also well assembled, while the *T. mercedesae* assembly is highly fragmented and with many gaps but has a reasonable scaffold N50. In themselves, these new resources are valuable in providing genomic information on mites that affect the commercially important honey bee *Apis mellifera*.

Beyond this, the authors pose an interesting scientific question as to the nature of evolutionary divergence of the *Varroa* sister species since their quite recent common ancestor, particularly given their high morphological similarity and capacity to exploit the same host species. The introduction presents a number of theoretical scenarios on the nature of host-parasite arms race dynamics, ectoparasite speciation, and the potential nature of selection in an ecosystem comprised of multiple, distinct ectoparasites within the same host.

However, determining relevant selective pressures that may affect the nature of the *Varroa* species' divergence is challenging, given that clear and thorough documentation of host diversity and mite geographic range is somewhat limited and conflated by *V. jacobsoni* having been a cryptic species. Clear mitochondrial haplotypes and genome assemblies in this study will clarify future work, but the current state of ecological knowledge is such that the discussion is highly speculative.

Moreover, while some attempt is made to characterize the *Varroa* species' genomes, I found the analyses somewhat limited and superficial, with the extent of genomic difference not sufficiently compelling to support the manuscript's title. A number of established pipelines are used to identify orthology clusters, tandem duplications, signatures of positive selection, and gene ontology enrichment. However, I missed any validation of these outputs, or a more nuanced use of GO terms for distinguishing the species biologically. For example, numerous highly conserved genes often occur in tandem clusters, including histones, cuticle structural proteins, and transcription factor and signaling molecules such as Hox genes. Tandem expansions are also typical for certain gene classes such as those that encode C2H2 zinc finger proteins and transposable elements (the latter potentially being consistent with GO terms mentioned in relation to endonuclease activity, but with no recognition or follow up on this). With this in mind, I am surprised by the very low number of reported tandem duplications, and the lack of inspection for expected candidates – here and for other pipeline outputs. For the six copies of the HSP70 protein in *V. destructor*, have the authors even checked that these are genuinely distinct genes and not an assembly artifact? A phylogenetic analysis would help. For genes under positive selection, it is interesting that there is little overlap in gene identity between the *Varroa* species, but the graphical and textual laundry list of selected GO terms is uninformative: there is no clear trend among those terms to flag a specific biological process, and the application of biological GO terms to distantly related species provides a very limited first approximation for actual *in vivo* function.

Lastly, although in general I could understand the results, the writing could be much stronger. There are numerous sentences that could be improved to be grammatically correct, particularly to avoid unintended scientific ambiguity, and this also includes the title of the paper. The second results section in particular is poorly written and poorly organized, with a continuous switching between

referring to *V. destructor* and *V. jacobsoni*. More generally, the presentation of the results would benefit from polishing to consider which information is truly relevant, and then to ensure that the reader is provided with sufficient explanatory text to make that relevance evident.

Reviewer #2 (Remarks to the Author):

In this manuscript, the authors present the high quality de novo genome assemblies of two sister species of Varroa mites. Both species infest and co-evolved with the eastern honey bees, but they recently shifted hosts to also infest the western honey bees, which causes honey bee decline and is of economic importance. The authors compare these Varroa mite genomes to search whether these two species evolved along similar evolutionary trajectories since their speciation event, or whether they show evidence of distinctive selective regimes that may have lead to divergence. The comparison of the genomes indicates that the genes that evolved under positive selection are mostly dissimilar between the two Varroa mite species, and that gene duplication events mostly affected different subsets of genes. These results are discussed in the context of the competitive interactions among Varroa mites species when they co-infect the same host (species), local adaptation and the evolutionary arms races with the host.

The comparison of the genomes of two parasitic sister species is very interesting, and addresses an original research question on divergent evolutionary trajectories for two closely related species in a very similar niche / host. The genome sequencing and resolving of the status of sister species is very convincing, and yielded high quality de novo assemblies for these two species. However, the comparison of the genomes is less convincing and clear to me. Also, the topics of adaptation to multispecies host-parasite interactions and divergent evolutionary trajectories, presented as the main focus of this manuscript, I consider to be the weakest aspect of this manuscript. Both the methodology and the results are not clearly presented, and also the concept of adaptation is not well worked out: the authors find a number of genes that show signatures of positive selection (although it is unclear to me exactly how they determine this), and a number of genes that have duplicated. They compare the GO categories for the genes under positive selection and the duplicated genes, and pronounce them both to be dissimilar between the Varroa species (although it is unclear to me exactly how they determine this). They interpret these genome difference as the consequence of interactions with the host, or between two competing parasites (although it is unclear to me on exactly what basis they consider this the most likely explanation), but unlikely to reflect local adaptation. Finally, throughout the manuscript the English writing could be improved, both to enhance clarity and to correct small errors.

Specific comments:

- Title: the title statement is too strongly formulated, as adaptation has not been investigated nor shown in this manuscript. Showing a signature of selection is not the same as showing an adaptation.
- Abstract: the focus on the abstract is mostly on the conceptual background of the research question and possible scenarios, while the description of the methods and primary results is unclear and presented only cursory. E.g., in L35-36 ("compare them to another mite") it is not clearly stated that the genomes are compared to the "genome" of another mite. The results are summarized in half a single sentence.
- Introduction: While the co-evolution and evolutionary history of the host-parasite interactions in Varroa mites and honey bees is presented well, I would appreciate a short description of the actual host-parasite interaction itself: what do Varroa mites do to their host and their broods, how do they inflict costs (in adults and broods), and what are the consequences in host fitness (mobility, morbidity, survival, fecundity) in the eastern and western honey bees?

- L120: the research question is not very clearly formulated: what "equilibrium" is meant here?
- Materials & methods: many aspects are not clear, either why they were done, or how they were done. For example, why were the mites collected from their new host, western honey bees, while the research question is how they arrived "at an equilibrium in the eastern honey bee"? Why was the X-ray micro-CT (L157) performed? Can you better explain the method that was used to test for genes that may have evolved under positive selection (L218-219)? How did you test for tandem duplications (L219-221)? How did you evaluate the semantic similarity (L225-227) in GO terms, and whether or not that was higher or lower than what you would expect by chance alone?
- Results: the methods for detecting "positive selection" are unclear from the methods, but from L376 - 379 it appears that the vast majority of these genes are lineage-restricted / novel genes? Is this correct, and if so, is this analysis of positive selection not confounded with lineage-specificity? Also, the overlap in these genes under positive selection between the Varroa species is substantial: I would consider an overlap of 40-50 out of ~230 not a trivial proportion (>20%), but here it is presented as "very different". Also, the overlap in semantic space (Fig 4) is interpreted as "no overlap", while it appears that there is considerable overlap. I don't understand how these plots are evaluated. In Fig 5A, only the duplications in *V. destructor* are presented. Why are the 16 duplications in *V. jacobsoni* not included in this graph? The conclusion in the legend "different genes and chromosomal regions" (L432-433) is not shown in this figure. Again, there seems to be some overlap in GO categories (Fig 5B), but it is concluded that there is no "substantial degrees of parallel evolution" (L436). Whether or not there is "overrepresentation" of particular GO categories depends both on the number of genes in each of the categories, the number of genes under positive selection, and the number of categories itself: there seems to be a fairly limited number of GO categories that contain genes with signatures of positive selection (out of the hundreds of possible GO categories), and several genes per category show these signatures in both species. These same aspects are also not included in Table 2, which makes it very difficult to evaluate whether or not the numbers of genes undergoing positive selection are more, less or equal to what you would expect by chance.
- Discussion: I am not fully convinced that the results fully support the main conclusion that the 2 Varroa mites have "undergone largely dissimilar evolutionary responses" (L477), and that this is "most likely driven by the ancestral host" (L480). I don't think that Figure 1 shows extensive mixing of the two Varroa species or range expansion: they still largely seem separated into two different geographic zones, with very limited overlap, while they mostly share the same host species? Why then is it concluded that the patterns reflect adaptation to the ancestral host, and not to the local environments? Is there any strong evidence for specificity/compatibility between host strains and parasite strains within or between geographic zones? Did anyone test for this specificity, or for local adaptation in common garden experiments? And may the parapatric speciation of the Varroa mites not reflect mostly genetic drift and differentiation, rather than competitive exclusion? Also, the sequencing was done on mites collected from western honey bees, yet the positive selection is considered to reflect the interaction with the ancestral host: these mites may have been adapting to their novel host, or to the acaricides that are frequently used for western honey bee hives, which could also result in signatures of positive selection. This should be discussed. Finally, the discussion is very speculative, with multiple scenarios to interpret the found patterns without experimental evidence or data to assess the likelihood of each scenario. The GO categories that were identified for the genes under positive selection or for duplicated genes do not relate in a straightforward manner to the various "adaptations" that are presented as possible explanations.

Minor / textual comments

- L98: reference is mis-formatted
- L148: "hive super" is jargon that I am unfamiliar with
- L152: "the" is missing in front of "hive"
- L245-247: unclear formulation
- L265: 25.4% improvement of what exactly?

- L358: word missing (after "enriched").
- L363: please, rephrase: it misses something on "criteria" or so in relation to the host ranges and the " had to comply" part.
- L367: "positively selected genes" is shorthand and not an entirely accurate description. Please, rephrase
- L368: it is not clear from this sentence whether the purple number (species-specific genes) is a subset of the green number, or only partially overlapping.
- L398: please add "under positive selection" after genes
- L511: which two species do you mean here?
- L534 -536: I don't quite understand how you come to this conclusion
- L560: "More Varroa [...] host 79,80". Sentence unclear and incomplete?

Dear Reviewers,

We would like to thank you for your time and effort in reviewing our manuscript. We appreciate your comments and we have made a number of changes to improve the manuscript, outlined in our point-by-point response below. We hope this new version will be clearer and more complete for future readers, and you will find our changes satisfactory.

Referee expertise:

Referee #1: Genome sequencing, comparative genomics in insects

Referee #2: Genomics of evolutionary adaptations

Reviewers' comments:

Reviewer #1 (Remarks to the Author):

This study presents a revised genome for the ectoparasitic mite *Varroa destructor* along with new draft assemblies for the recently recognized distinct species *V. jacobsoni* and the more distantly related *Tropilaelaps mercedesae*. The new assembly for *V. destructor* has high contiguity and achieves nearly chromosomal-level assembly, although claims about it being the best assembled arthropod genome are unneeded, likely to be quickly superseded, and do not seem to consider the genomes of the insects *Drosophila* or *Tribolium*. The *V. jacobsoni* genome is also well assembled, while the *T. mercedesae* assembly is highly fragmented and with many gaps but has a reasonable scaffold N50. In themselves, these new resources are valuable in providing genomic information on mites that affect the commercially important honey bee *Apis mellifera*.

Reply:

The comments about genome quality and contiguity ranking are totally on a point and we corrected this in the manuscript and abstract to point out that our *Varroa* genomes are among the best of Arachnids and Mesostigmata. We corrected this in our abstract as well as in the other section.

We also notice that we may have confused readers in the abstract on the use of *Tropilaelaps mercedesae* genome. This genome was previously published from the study of Dong et al. (2017), Gigascience and we simply used it for comparison with our *Varroa* genomes. For this reason, we corrected the abstract and tried to be more precise about the origins of published genomes used for comparative genomics. We also removed this genome from Table 1 to minimize confusion.

Beyond this, the authors pose an interesting scientific question as to the nature of evolutionary divergence of the *Varroa* sister species since their quite recent common ancestor, particularly given their high morphological similarity and capacity to exploit the same host species. The introduction presents a number of theoretical scenarios on the nature of host-parasite arms race dynamics, ectoparasite speciation, and the potential nature of selection in an ecosystem comprised of multiple, distinct ectoparasites within the same host.

However, determining relevant selective pressures that may affect the nature of the Varroa species' divergence is challenging, given that clear and thorough documentation of host diversity and mite geographic range is somewhat limited and conflated by *V. jacobsoni* having been a cryptic species. Clear mitochondrial haplotypes and genome assemblies in this study will clarify future work, but the current state of ecological knowledge is such that the discussion is highly speculative.

Reply:

We agree that there is still a lot of uncertainty existing regarding Varroa ecology, distribution and the exact nature with their original host *A. cerana* in Asia since the genus is highly cryptic. *V. destructor* and *V. jacobsoni* were mistaken as the same species until 2000 and most reports on their presence in Asia was done before that key time. Some previous studies attempted to draw their possible geographical distribution and exact host range (e.g. Chantanakul et al. 2015) but were considering all reports without necessary molecular verification. Here, we reduce assumptions on both species distribution and host range, by reviewing Varroa literature and keeping only data points confirmed by mtDNA barcoding. We also included other Varroa species such as *V. underwoodi* and *Varroa sp.* presented on *A. cerana* to have an overview of the genus and species overlap in the native area.

All data used for the species range are available on the associated GitHub for this paper <https://maevatecher.github.io/varroa-denovo-genomes/#1> reported distribution of varroa mites on the eastern honey bee, and we also provided an interactive map showing each mtDNA haplotype described for *V. destructor* and *V. jacobsoni*. We hope this additional online resource will be useful for future population genetics and taxonomic studies.

Therefore, for the purposes of this study, we believe that we addressed the reviewer's concerns about taxonomy and how it relates to ecology.

Moreover, while some attempt is made to characterize the Varroa species' genomes, I found the analyses somewhat limited and superficial, with the extent of genomic difference not sufficiently compelling to support the manuscript's title. A number of established pipelines are used to identify orthology clusters, tandem duplications, signatures of positive selection, and gene ontology enrichment. However, I missed any validation of these outputs, or a more nuanced use of GO terms for distinguishing the species biologically.

As discussed below (and also in response to the other reviewer), we decided against over-interpreting the GO term results, because we could not discern obvious major tendencies in the data.

For example, numerous highly conserved genes often occur in tandem clusters, including histones, cuticle structural proteins, and transcription factor and signaling molecules such as Hox genes. Tandem expansions are also typical for certain gene classes such as those that encode C2H2 zinc finger proteins and transposable elements (the latter potentially being consistent with GO terms mentioned in relation to endonuclease activity, but with no recognition or follow up on this). With this in mind, I am surprised by the very low number of reported tandem duplications, and the lack of inspection for expected candidates – here and for other pipeline outputs.

For the six copies of the HSP70 protein in *V. destructor*, have the authors even checked that these are genuinely distinct genes and not an assembly artifact? A phylogenetic analysis would help.

Reply:

We agree that a phylogenetic analysis would have been helpful from the start for the reader and we implemented this in the new draft as an online material https://maevatecher.github.io/varroa-denovo-genomes/#similarity_of_duplicated_genes_and_trees

We manually checked all duplication events detected by MSOAR2 pipeline especially when more than two genes were involved. For that, we aligned the coding sequence of each duplicated gene from the same cluster using ClustalW and making trees with IQ-TREE. We validated all duplications excepted one in the *V. destructor* Histone 2A group. As detailed under the Histone tree we explained “It seems that the CDS region for the two loci LOC111250934 on chromosome 1 and LOC111255102 on the contig BEIS01000444.1 are 100% identical. Since the contig BEIS01000444.1 could result from assembly artifact and in a conservative way, we prefer to consider that only LOC111250934 and LOC111250948 should be considered here.”

For genes under positive selection, it is interesting that there is little overlap in gene identity between the *Varroa* species, but the graphical and textual laundry list of selected GO terms is uninformative: there is no clear trend among those terms to flag a specific biological process, and the application of biological GO terms to distantly related species provides a very limited first approximation for actual in vivo function.

We agree with the reviewer that there do not seem to be obvious central biological tendencies in species-specific GO terms or their overlaps. As a result, we tried to avoid over-interpreting the GO-term based results and present them mainly in the supplement, with the exception of a figure illustrating the poor overlap in gene ontology categories. Instead, we focus on other findings in the main text.

Additionally, we realized that the previous figure 4B did not help visualizing the overlap level between GO terms (only 0.7% (3/(201+190))). Thus we redraw the scatterplot from REVIGO by removing the GO term associated with *Varroa* ancestral genes under positive selection (before sibling species split). That way we found it is easier to see that with a small similarity Resnik’s measure, the GO term overlap areas were very local and concerned cellular responses to external factors (alcohol, UV, lipid, sterol,...) and floor plate development (floor plate, dorsal opening, head involution,...).

Lastly, although in general I could understand the results, the writing could be much stronger. There are numerous sentences that could be improved to be grammatically correct, particularly to avoid unintended scientific ambiguity, and this also includes the title of the paper. The second results section in particular is poorly written and poorly organized, with a continuous switching between referring to *V. destructor* and *V. jacobsoni*. More generally, the presentation of the results would

benefit from polishing to consider which information is truly relevant, and then to ensure that the reader is provided with sufficient explanatory text to make that relevance evident.

Reply:

The revised manuscript has been reviewed and corrected by a technical editor. We hope that this new version will be more concise and bring more clarity in particular on the results.

Reviewer #2 (Remarks to the Author):

In this manuscript, the authors present the high quality de novo genome assemblies of two sister species of Varroa mites. Both species infest and co-evolved with the eastern honey bees, but they recently shifted hosts to also infest the western honey bees, which causes honey bee decline and is of economic importance. The authors compare these Varroa mite genomes to search whether these two species evolved along similar evolutionary trajectories since their speciation event, or whether they show evidence of distinctive selective regimes that may have lead to divergence. The comparison of the genomes indicates that the genes that evolved under positive selection are mostly dissimilar between the two Varroa mite species, and that gene duplication events mostly affected different subsets of genes. These results are discussed in the context of the competitive interactions among Varroa mites species when they co-infect the same host (species), local adaptation and the evolutionary arms races with the host.

The comparison of the genomes of two parasitic sister species is very interesting, and addresses an original research question on divergent evolutionary trajectories for two closely related species in a very similar niche / host. The genome sequencing and resolving of the status of sister species is very convincing, and yielded high quality de novo assemblies for these two species. However, the comparison of the genomes is less convincing and clear to me. Also, the topics of adaptation to multispecies host-parasite interactions and divergent evolutionary trajectories, presented as the main focus of this manuscript, I consider to be the weakest aspect of this manuscript. Both the methodology and the results are not clearly presented, and also the concept of adaptation is not well worked out: the authors find a number of genes that show signatures of positive selection (although it is unclear to me exactly how they determine this), and a number of genes that have duplicated. They compare the GO categories for the genes under positive selection and the duplicated genes, and pronounce them both to be dissimilar between the Varroa species (although it is unclear to me exactly how they determine this). They interpret these genome difference as the consequence of interactions with the host, or between two competing parasites (although it is unclear to me on exactly what basis they consider this the most likely explanation), but unlikely to reflect local adaptation. Finally, throughout the manuscript the English writing could be improved, both to enhance clarity and to correct small errors.

Specific comments:

– Title: the title statement is too strongly formulated, as adaptation has not been investigated nor shown in this manuscript. Showing a signature of selection is not the same as showing an adaptation.

Reply:

We changed the title to “Divergent selection following speciation in two ectoparasitic honey bee mites”

We implemented additional analysis into the manuscript to further explore the idea of host adaptation by looking at chemosensory proteins (OBP, SNMP, NPC2, IGR, and GR) as well as the ecdysteroid pathway with cytochrome p450 genes involved in oogenesis. We choose chemosensory receptors and soluble carrier proteins families, as Varroa mites are blind and use their chemosensory system (forelegs) to find their bee host and communicate. Two previous transcriptomics studies on Varroa mite females (Eliash et al. 2017, 2018) have identified and listed proteins expressed in their sensory organs (forelegs). We used the protein sequences to BLAST against our Varroa genomes 1) to improve this repertoire and 2) to compare both mite adaptive strategies especially as no report existed for *V. jacobsoni*.

Regarding the ecdysteroid pathway, the absence of four Halloween genes (Cabrera et al. 2015) involved in ecdysone biosynthesis, have been previously reported in *V. destructor*. By looking at the phylogeny of the predicted proteins, we could show that although chemosensory genes and cytochrome p450 genes family are mostly conserved, non-negligible differences appeared between both sibling species. Additionally, we found more homologs of Halloween genes in our new Varroa assemblies genomes, missed by a previous study using 454 pyrosequencing (Cabrera et al. 2015). Thus in our phylogenetic analysis (Figure 7), we found also *shadow* and similar proteins to *shroud* and *neverland*.

Given our new outlook on the adaptation part, we decided to reformulate the title but keeping the original idea.

– Abstract: the focus on the abstract is mostly on the conceptual background of the research question and possible scenarios, while the description of the methods and primary results is unclear and presented only cursory. E.g., in L35-36 ("compare them to another mite") it is not clearly stated that the genomes are compared to the "genome" of another mite. The results are summarized in half a single sentence.

Reply:

We tried to be more precise regarding the use and comparison with other genomes and added as many details as 150 words limit allowed us without being confusing.

– Introduction: While the co-evolution and evolutionary history of the host-parasite interactions in Varroa mites and honey bees is presented well, I would appreciate a short description of the actual host-parasite interaction itself: what do Varroa mites do to their host and their broods, how do they inflict costs (in adults and broods), and what are the consequences in host fitness (mobility, morbidity, survival, fecundity) in the eastern and western honey bees?

– L120: the research question is not very clearly formulated: what "equilibrium" is meant here?

Reply:

We realized that such details about Varroa mites and honey bee host interactions were missing in the introduction and we corrected that by reviewing entirely it. We hope that the costs associated with Varroa parasitism and exact nature of adaptation and counter-adaptation will much clearer here.

With this new version of the introduction, we specified what was meant by tolerant “equilibrium” by showing the balanced relationship maintained between *A. cerana* and Varroa mites contrary to the unbalanced relationship between *A. mellifera* and *V. destructor*. We also emphasize that those differences in “equilibrium” come from the more recent coevolutionary history between the mites and the Western honey bee which was a naive host to any brood ectoparasitic mites until 80 years ago.

– Materials & methods: many aspects are not clear, either why they were done, or how they were done. For example, why were the mites collected from their new host, western honey bees, while the research question is how they arrived "at an equilibrium in the eastern honey bee"?

Reply:

Given the lack of clarity pointed out here, we corrected and reviewed the Material and Methods section and hope this version will be clearer regarding our sequencing strategy choice and sampling chosen to respond adequately.

Although it would have been ideal to sequence mites from the original host to strictly address directly the question of evolution and adaptation to the eastern honey bee, we choose to collect mites from the western honey bee. One of the considerations for doing so the large amount of DNA necessary for the PacBio RSII platform. It is not feasible to obtain a sufficient number of mites from the native host. Secondly, eastern honey bee colonies are more difficult to put on frames than *A. mellifera*, and collection of a brood parasite is very invasive and can be destructive. As we had access to our experimental apiary of *A. mellifera* with the infestation of Varroa all year long (tropical environment of Okinawa), we could collect enough mites for long-range sequencing without destroying the sampled colonies. Finally, since we sequenced *V. destructor* from *A. mellifera* and to reduce bias for comparison, we choose to sequence *V. jacobsoni* from *A. mellifera* (from Papua New Guinea, where its host switch was detected in 2008).

Why was the X-ray micro-CT (L157) performed?

Reply:

We used here an X-ray micro CT for illustrative purposes to show that both cryptic species are very difficult to differentiate for non-Acari or honey bee experts. If many illustrations such as electron microscopic images or pictures exist for *V. destructor*, the latest image available for *V. jacobsoni* was published in the revision study of Anderson and Trueman in 2000 which defined these species as two separate entities. We also believed that a visual association of the model studied can always be a plus for species/genus centered studies. In the future, we hope to make available the scan data of both mites which can be used for morphological studies at the Varroa scale or even Mesostigmata.

Can you better explain the method that was used to test for genes that may have evolved under positive selection (L218-219)? How did you test for tandem duplications (L219-221)?

Reply:

For every orthogroup predicted by Orthonome with >3 species - gene trees were constructed and then reconciled using the species tree. The branches of this tree were then tested for positive selection using aBSREL (adaptive Branch-Site Random Effects Likelihood, from the HyPhy suite), which is an improved version of the commonly-used "branch-site" models such as codeml. Tests were FDR corrected and only branches with a significant p-value (<0.05) were considered to be positively selected in each gene.

Orthonome predicts orthogroups and their respective duplications using tree reconciliation and graph resolution. This leverages duplications predicted by MSOAR2 (Shi et al. 2010) between pairs of species. We then identified species-specific duplications (with respect to all lineages analyzed) as genes which were tagged as 'duplicates' by Orthonome, had the highest similarity (Smith Waterman) to an orthologue, and were immediately adjacent to the same gene. While this is a conservative approach - it allows us to use all available evidence to make the most confident prediction of tandem duplications.

We added in the online supplemental data the raw output from HyPhy regarding genes under positive selection for each branch:

<https://maevatecher.github.io/varroa-denovo-genomes/>

How did you evaluate the semantic similarity (L225-227) in GO terms, and whether or not that was higher or lower than what you would expect by chance alone?

Reply:

We apologize for our oversight on this method details. We forgot to specify that the semantic similarity was measured using the SimRel option (by default in REVIGO) which is also known as the Resnik's measure. Resnik, J Artif Intell Res (1999), 11:95-130. We added this detail in the manuscript.

Both reviewers pointed out that we neglected to test for the significance of overlap. We performed Fisher's exact tests and the overlap between genes under positive selection for *V. destructor* (n= 234) and *V. jacobsoni* (n= 225) was significant considering that we had 9,628 orthologous genes. However, looking at the GO terms, we found that the overlap was not significant indicating that the 201 and 190 GO terms enriched for the positively selected genes of both mites are involved in divergent biological processes, molecular functions, and cellular components.

More details and code here: https://maevatecher.github.io/varroa-denovo-genomes/#fisher%E2%80%99s_exact_test_on_overlapping_genes

- Results: the methods for detecting "positive selection" are unclear from the methods, but from L376 - 379 it appears that the vast majority of these genes are lineage-restricted / novel genes?

Is this correct, and if so, is this analysis of positive selection not confounded with lineage-specificity?

Reply:

Only orthogroups with >3 species out of 5 were tested to avoid phylogenetic bias in our tests. The resulting branches were then tested for positive selection as aforementioned. There would hence be no lineage specificity in these tests at the risk of losing some lineage-specific signal - which we have attempted to test for using lineage-specific orthogroups. The methods used for detecting selection have been expanded.

Also, the overlap in these genes under positive selection between the *Varroa* species is substantial: I would consider an overlap of 40-50 out of ~230 not a trivial proportion (>20%), but here it is presented as "very different". Also, the overlap in semantic space (Fig 4) is interpreted as "no overlap", while it appears that there is considerable overlap. I don't understand how these plots are evaluated. In Fig 5A, only the duplications in *V. destructor* are presented. Why are the 16 duplications in *V. jacobsoni* not included in this graph?

Reply:

Though the exact number of genes were listed in Figure 3, we apologize that the text was not sufficiently clear. Of the total ~230 genes under positive selection in each species, only 13 genes overlap ($13/(225+234) \sim 2.8\%$). There were an additional 40 genes that were under selection on the ancestral branch leading up to the split between the two *Varroa* species. These correspond to selection pressures before the two species split and don't represent species-specific selection regimes. We have addressed the overlap questions explicitly using exact tests. Interestingly, this number is higher than one would expect by chance, but hardly an overwhelming sign of convergence. We added a more detailed and clearer presentation of the overlaps to the results.

Regarding duplications on Fig5 for *V. jacobsoni*. some of the genes involved did not have orthologs identified by ORTHONOME with *V. destructor* which made the plot assuming synteny incomplete. However, we made available a similar karyogram-like for *V. jacobsoni* scaffolds showing that most duplication is in tandem on adjacent regions except for the duplication of the ABC gene family (see details here <https://maevatecher.github.io/varroa-denovo-genomes/>).

The conclusion in the legend "different genes and chromosomal regions" (L432-433) is not shown in this figure. Again, there seems to be some overlap in GO categories (Fig 5B), but it is concluded that there is no "substantial degrees of parallel evolution" (L436). Whether or not there is "overrepresentation" of particular GO categories depends both on the number of genes in each of the categories, the number of genes under positive selection, and the number of categories itself: there seems to be a fairly limited number of GO categories that contain genes with signatures of positive selection (out of the hundreds of possible GO categories), and several genes per category show these signatures in both species. These same aspects are also not included in Table 2, which makes it very difficult to evaluate whether or not the numbers of genes undergoing positive selection are more, less or equal to what you would expect by chance.

We have added a test of whether GO term categories overlap more than expected by chance (answer: no), to help quantify this argument available in the varroa-denovo-genomes GitHub R markdown.

– Discussion: I am not fully convinced that the results fully support the main conclusion that the 2 *Varroa* mites have "undergone largely dissimilar evolutionary responses" (L477), and that this is "most likely driven by the ancestral host" (L480). I don't think that Figure 1 shows extensive mixing of the two *Varroa* species or range expansion: they still largely seem separated into two different geographic zones, with very limited overlap, while they mostly share the same host species?

Reply:

We agree with the comments that our results do not support as important dissimilar evolutionary responses as our writing suggested, so we rewrote it in a more nuance way. The revised discussion is also pointed out now "Differences between the two species most likely result from local adaptation to host geographic subspecies, the environment or their interaction." Indeed, according to mtDNA, both mites distribution appeared parapatric on *A. cerana* and we agree that the contact area seems restricted to South Asia. When we indicate large overlap we referred to the idea that *Varroa destructor* is now a cosmopolitan species and overlap *A. mellifera* distribution following host switch and range expansion.

Why then is it concluded that the patterns reflect adaptation to the ancestral host, and not to the local environments?

Reply:

We apologize if the conclusions appeared misleading. We don't believe that it's possible to distinguish adaptations to the host vs. the environment using our data alone, because they are not independent factors. We can only speculate about the possible causes of the different selection pressures and have made this point clearer. There is independent evidence that *A. cerana* geographic races area associated with distinct *Varroa* lineages, suggesting host adaptation, and we added this to the discussion.

Is there any strong evidence for specificity/compatibility between host strains and parasite strains within or between geographic zones? Did anyone test for this specificity, or for local adaptation in common garden experiments?

Reply:

This is an excellent point that we would have like to know but unfortunately, such data do not exist to the best of our knowledge. One interesting fact is that *V. destructor* and *V. jacobsoni* have been shown to infect the same host *A. cerana* genetic population in Thailand (parapatric region) in Warrit et al. (2006) *Apidologie*. A later study on the same dataset by Rueppell et al. (2011) showed that the hypothesis of local host-pathogen coevolution was not supported by microsatellites data but that observed structure would be rather shaped by biogeographic history and migration. We hope that in the future local adaptation for each species will be investigated in common garden experiments which should be possible in Thailand mainly according to the distribution reported by our *Varroa* mtDNA map.

One evidence of host specificity for Varroa has been reported in the Philippines and Vietnam for *V. destructor* strains and another Varroa sp. in which certain strains will only infect *A. mellifera* (Korean and LB) while others strains will infect the main host *A. cerana* in Beaurepaire et al. (2015), Plos One. To our knowledge, no such host specificity or compatibility has been reported for *V. jacobsoni*.

And may the parapatric speciation of the Varroa mites not reflect mostly genetic drift and differentiation, rather than competitive exclusion?

The reviewer brings up a good point – if everything is neutral, we would also expect little overlap. Genetic drift may indeed explain the duplication data, since we don't have a good model to test whether duplications are adaptive or neutral. All we can say there is that not the same genes have undergone duplication. However, the null model for the random effects branch-site model used in the analysis is neutral evolution, i.e. drift, or negative selection. So, if we are to assume that these tests work, the selection is acting on something, just the genes involved are different in the two species.

Also, the sequencing was done on mites collected from western honey bees, yet the positive selection is considered to reflect the interaction with the ancestral host: these mites may have been adapting to their novel host, or to the acaricides that are frequently used for western honey bee hives, which could also result in signatures of positive selection. This should be discussed.

This paragraph was re-written to make it more nuanced. In reality, only *V. destructor* has been extensively exposed to miticides. So, most of the genes in the *V. jacobsoni* genome likely result from more ancient selection.

Following on a suggestion from Reviewer 1, we expanded the analysis of genes that should be involved in host-specific adaptations, including odorant binding proteins, gustatory receptors and chemosensory receptors and others, and identified patterns of species-specific selection. There is now an extensive supplement with this analysis, providing at least some evidence for host-specific evolution.

Finally, the discussion is very speculative, with multiple scenarios to interpret the found patterns without experimental evidence or data to assess the likelihood of each scenario.

We re-wrote the discussion to emphasize that many aspects of the mites' biology are not well known and to identify areas where future research is needed. In essence, we conclude that the patterns of evolution are not the same in these closely related mites, and outline reasons why this might be the case. The aim of the discussion is summarized in its opening paragraph: "Though not enough data exist to distinguish between these alternatives, below we summarize what is already known and outline promising avenues for investigation".

The GO categories that were identified for the genes under positive selection or for duplicated genes do not relate in a straightforward manner to the various "adaptations" that are presented as possible explanations.

Given the relatively small number of genes under selection GO term analysis may lack power to identify important trends, because many of the genes will also lack proper annotation. Therefore, we have been weary against over-interpreting the GO term results, and have done so sparingly.

Minor / textual comments

- L98: reference is mis-formatted
- L148: "hive super" is jargon that I am unfamiliar with
- L152: "the" is missing in front of "hive"
- L245-247: unclear formulation
- L265: 25.4% improvement of what exactly?
- L358: word missing (after "enriched").
- L363: please, rephrase: it misses something on "criteria" or so in relation to the host ranges and the " had to comply" part.
- L367: "positively selected genes" is shorthand and not an entirely accurate description. Please, rephrase
- L368: it is not clear from this sentence whether the purple number (species-specific genes) is a subset of the green number, or only partially overlapping.
- L398: please add "under positive selection" after genes
- L511: which two species do you mean here?
- L534 -536: I don't quite understand how you come to this conclusion
- L560: "More Varroa [...] host 79,80". Sentence unclear and incomplete?

Reply:

All minor comments were corrected accordingly.

REVIEWERS' COMMENTS:

Reviewer #2 (Remarks to the Author):

The current version of the manuscript is much improved in terms of clarity and nuance. The additional analyses and descriptions of methods, as well as the textual revisions provide a much stronger argumentation and comparison of the two genomes, while the focus is also clearer. The manuscript is now more convincing in showing divergent patterns and evolutionary trajectories in/from the genomes of the two *Varroa* species, with more clarity and detail on the ecological and evolutionary context.

I have a few remaining queries for the authors, primarily on the introduction which is still unclear in some parts. There are several sentences that are confusing or even contradictory. Also, the research question is not accurately formulated (see below for details).

In addition to these queries for the introduction, I have a remaining query on the methods. If I understand it correctly, only a single haploid male was sequenced for *V. jacobsoni*. This is used for de novo genome construction (which is not necessarily problematic), but also for identifying bi-allelic SNPs (line 365). The latter requires more explanation, and also some caution when used to compare with *V. destructor* for which many more individuals were sequenced.

- title:

The new title is much more accurate than the previous one, although I still have a small query: why do the authors call it "divergent selection", rather than "divergent evolution" or "divergent evolutionary trajectories"? How can we know whether selection was different, when the authors argue that the two species evolved within the same niche (*A. cerana*), and that external conditions in the phoretic stage do not appear to be prohibitive to survival, which could suggest very similar selection pressures? Perhaps the selection was largely similar, but the evolutionary responses divergent?

- introduction: there are still some unclear, confusing or contradictory sentences in the introduction. The paragraph on co-evolution between the *Varroa* mites and honey bees (from line 93 onwards) is still confusing and partly unclear:

- The first statement (The dramatic effects ... are due to their recent coevolutionary history) is too strongly formulated: how can we be sure that it is indeed their recent coevolutionary history that causes these dramatic effects? Moreover, I think the authors wish to claim/suggest that it is a lack of co-evolutionary history that is making the bees so susceptible. This needs rephrasing, and much more careful wording.
- The next sentence is also confusing and partly unclear: "Originally, brood parasitism [...] has evolved in all *Apis* species except for *A. mellifera*." I expect that brood parasitism evolved once in the ancestral species (of *Varroa* mites and bees), and was then secondarily lost in *A. mellifera*? What is known about when the association between *Varroa* mites and *Apis* started, and why it did not evolve/got lost in *A. mellifera*?
- Then the description of *V. jacobsoni* and *V. destructor* is incomplete. It is explained that these were originally considered a single species. It is, however, not explained, when / why / how they were separated in 2 species.
- The research question is at odds with the earlier statements and not accurate, and needs adjusting: "Here, we ask how both mite species have arrived at this tolerance equilibrium" (line 116). Firstly, only one mite species is considered to be in tolerance, whereas the *A. mellifera* is considered to be not to be (line 100). Secondly, this is not what the study is doing. It is not comparing tolerance and how it got there, but comparing genomes.
- The statement in line 122 ("*V. jacobsoni*, which does not parasitize *A. mellifera* in this region (=south east asia) is in contradiction with the text of the legend of figure 1 ("*V. jacobsoni* [...] extended its

range into Papua New Guinea [...] followed by a shift to *A. mellifera*)

Minor comments:

line 52: add "the" before "recognized animal phyla"

line 62: word missing after heterospecific (e.g. parasites?)

line 64: I would suggest to delete "For example"

line 85: the first part of this sentence needs rephrasing: "Parasitic feeding [...] abnormal development"

line 11: replace "burst" with other word, e.g. "boosted" or "facilitated"

REVIEWERS' COMMENTS:

Reviewer #2 (Remarks to the Author):

Dear Reviewer,

We would like to thank you once again for your time and comments. We hope we could clear the last questions regarding the SNPs calling and we corrected minor writing errors according to your comments. The section of the introduction describing the interaction of Varroa mites and honey bee has also been rewritten to reduce ambiguities. We hope you find these improvements acceptable.

The current version of the manuscript is much improved in terms of clarity and nuance. The additional analyses and descriptions of methods, as well as the textual revisions provide a much stronger argumentation and comparison of the two genomes, while the focus is also clearer. The manuscript is now more convincing in showing divergent patterns and evolutionary trajectories in/from the genomes of the two Varroa species, with more clarity and detail on the ecological and evolutionary context.

I have a few remaining queries for the authors, primarily on the introduction which is still unclear in some parts. There are several sentences that are confusing or even contradictory. Also, the research question is not accurately formulated (see below for details).

In addition to these queries for the introduction, I have a remaining query on the methods. If I understand it correctly, only a single haploid male was sequenced for *V. jacobsoni*. This is used for de novo genome construction (which is not necessarily problematic), but also for identifying bi-allelic SNPs (line 365). The latter requires more explanation, and also some caution when used to compare with *V. destructor* for which many more individuals were sequenced.

Reply:

Actually, for consistency, variant calls for both genomes were based on Illumina from single haploid males mapped to the scaffolded *destructor* reference genome. In this way there should be no bias, since we are using the *destructor* genome as a common set of coordinates only. Because sequence divergence is ~ 0.38% (over the 368,942,295bp) there should be minimal reference-induced bias. We re-wrote this section to clarify the methods and eliminate potential for confusion.

- title:

The new title is much more accurate than the previous one, although I still have a small query: why do the authors call it "divergent selection", rather than "divergent evolution" or "divergent evolutionary trajectories"? How can we know whether selection was different, when the authors argue that the two species evolved within the same niche (*A. cerana*), and that external conditions in the phoretic stage do not appear to be prohibitive to survival,

which could suggest very similar selection pressures? Perhaps the selection was largely similar, but the evolutionary responses divergent?

Reply:

We agree with the point raised here regarding our words ambiguity regarding selection process so we decided to change the title “divergent selection” by the more accurate “divergent evolutionary trajectories” as suggested.

- introduction: there are still some unclear, confusing or contradictory sentences in the introduction.

The paragraph on co-evolution between the Varroa mites and honey bees (from line 93 onwards) is still confusing and partly unclear:

- The first statement (The dramatic effects ... are due to their recent coevolutionary history) is too strongly formulated: how can we be sure that it is indeed their recent coevolutionary history that causes these dramatic effects? Moreover, I think the authors wish to claim/suggest that it is a lack of co-evolutionary history that is making the bees so susceptible. This needs rephrasing, and much more careful wording.
- The next sentence is also confusing and partly unclear: "Originally, brood parasitism [...] has evolved in all Apis species except for *A. mellifera*." I expect that brood parasitism evolved once in the ancestral species (of Varroa mites and bees), and was then secondarily lost in *A. mellifera*? What is known about when the association between Varroa mites and Apis started, and why it did not evolve/got lost in *A. mellifera*?
- Then the description of *V. jacobsoni* and *V. destructor* is incomplete. It is explained that these were originally considered a single species. It is, however, not explained, when / why / how they were separated in 2 species.

Reply:

We understand that some statements about mite-bee evolution may have been too strong in in the introduction. We clarified and rewrote the section following the historical description and invasion of Varroa mites worldwide. We hope this version will be more accurate and avoids confusion.

- The research question is at odds with the earlier statements and not accurate, and needs adjusting: "Here, we ask how both mite species have arrived at this tolerance equilibrium" (line 116). Firstly, only one mite species is considered to be in tolerance, whereas the *A. mellifera* is considered to be not to be (line 100). Secondly, this is not what the study is doing. It is not comparing tolerance and how it got there, but comparing genomes.

Reply:

The research question addressed here referred to the tolerance of both mite species by *A. cerana*. We rephrased the question focusing on evolution up to the present day “we ask how both mite species evolved toward a tolerance equilibrium with *A. cerana*”. The rest of the paragraph sets up alternative hypotheses and sets up genome sequencing as the tool to address them. We hope these adjustments made the framing clearer.

- The statement in line 122 ("V.jacobsoni, which does not parasitize A. mellifera in this region (=south east asia) is in contradiction with the text of the legend of figure 1 ("V. jacobsoni [...] extended its range into Papua New Guinea [...] followed by a shift to A. mellifera)

Reply:

We see the point here, and it might arise from our perception of South East Asia. We considered the Papua New Guinea region to be part of Oceania and not Asia. To avoid confusion we added "V. jacobsoni, which does not parasitize A. mellifera outside of Melanesia".

Minor comments:

line 52: add "the" before "recognized animal phyla"

line 62: word missing after heterospecific (e.g. parasites?)

line 64: I would suggest to delete "For example"

line 85: the first part of this sentence needs rephrasing: "Parasitic feeding [...] abnormal development

line 11: replace "burst" with other word, e.g. "boosted" or "facilitated"

Reply: We appreciate these minor corrections and added them to the manuscript as recommended.